

# The simulated climate of the Last Glacial Maximum and insights into the global carbon cycle

Pearse J. Buchanan[1], Richard J. Matear[2], Andrew Lenton[2], Steven J. Phipps[1], Zanna Chase[1], and David M. Etheridge[3]

[1]Institute for Marine and Antarctic Studies, University of Tasmania, Hobart, Tasmania, Australia
[2]CSIRO Oceans and Atmosphere, CSIRO Marine Laboratories, G.P.O Box 1538, Hobart, Tasmania, Australia
[3]CSIRO Marine Research, Aspendale, Victoria, Australia

*Correspondence to:* Pearse James Buchanan (pearse.buchanan@utas.edu.au)

**Abstract.** The ocean's ability to store large quantities of carbon, combined with the millennial longevity over which this reservoir is overturned, has implicated the ocean as a key driver of glacial-interglacial climates. However, the combination of processes that cause an accumulation of carbon within the ocean during glacial periods is still under debate. Here we present simulations of the Last Glacial Maximum (LGM) using the CSIRO Mk3L-COAL Earth System Model to test the contribution

of physical and biogeochemical processes to ocean carbon storage. For the LGM simulation, we find a significant global cooling of the surface ocean (3.2 °C) and the expansion of both minimum (Northern Hemisphere: 105 %; Southern Hemisphere: 225 %) and maximum (Northern Hemisphere: 145 %; Southern Hemisphere: 120 %) sea ice cover broadly consistent with proxy reconstructions. Within the ocean, a significant reorganisation of the large-scale circulation and biogeochemical fields occurs. The LGM simulation stores an additional 322 Pg C in the deep ocean relative to the Pre-Industrial (PI) simulation, particularly

due to a strengthening in Antarctic Bottom Water circulation. However, 839 Pg C is lost from the upper ocean via equilibration with a lower atmospheric $CO_2$ concentration, causing a net loss of 517 Pg C relative to the PI simulation. The LGM deep ocean also experiences an oxygenation (>100 mmol $O_2$ m$^{-3}$) and deepening of the aragonite saturation depth (> 2,000 m deeper) at odds with proxy reconstructions. Hence, physical changes cannot in isolation produce plausible biogeochemistry nor the required drawdown of atmospheric $CO_2$ of 80-100 ppm at the LGM. With modifications to key biogeochemical processes, which

include an increased export of organic matter due to a simulated release from iron limitation, a deepening of remineralisation and decreased inorganic carbon export driven by cooler temperatures, we find that the carbon content in the glacial oceanic reservoir can be increased (326 Pg C) to a level that is sufficient to explain the reduction in atmospheric and terrestrial carbon at the LGM (520 ± 400 Pg C). These modifications also go some way to reconcile simulated export production, aragonite saturation state and oxygen fields with those that have been reconstructed by proxy measurements, thereby implicating changes

in ocean biogeochemistry as an essential driver of the climate system.

Keywords: atmospheric $CO_2$, glacial-interglacial cycles, palaeoclimate modelling, ocean biogeochemical cycles, Climate System Model





# 1 Introduction

The late Pleistocene is characterised by a sawtooth-like cycling between cool glacial and warm interglacial states (Emiliani, 1966; Shackleton, 1967). Global temperatures and atmospheric $CO_2$ are strongly correlated across these climate cycles with approximately 80-100 ppm of change corresponding to global-mean temperature variations of 3-4 °C (Grootes and Stuiver,
1997; Jouzel et al., 1987; Parrenin et al., 2013; Petit et al., 1999). This correlation provides an important clue that aids our understanding of how the Earth experiences periods of warm and cold climate. Given the larger carbon storage potential of the ocean compared to the land and atmosphere, and given that major changes to oceanic circulation and productivity occur over multi-millennial timescales, it is now widely acknowledged that the ocean is a major player in driving glacial-interglacial changes in atmospheric $CO_2$ (Skinner et al., 2015; Wilson et al., 2015; Yu et al., 2014). However, identifying the combination
of mechanisms that drove a flux of carbon into the ocean at the LGM remains a fundamental and largely unresolved problem.

If we first consider only physical changes, a net influx of $CO_2$ caused by cooling is a feature of the glacial ocean. However, the influx attributed to cooling is partially offset by increased salinity in the glacial ocean, so that the total magnitude of influence by cooling is small and constrained to roughly 15 ppm (Brovkin et al., 2007; Menviel et al., 2012; Sigman and
Boyle, 2000). Therefore, other physical changes that partition more carbon in the deep ocean, notably changes to the large-scale circulation and sea-ice fields, may make a considerable contribution. Substantial research effort has revealed that the glacial circulation is indeed conducive to storing more carbon in the ocean, with a greater proportion of the deep ocean dominated by southern source waters (Adkins, 2013; Duplessy et al., 1988; Oliver et al., 2010; Skinner et al., 2010; Watson and Naveira Garabato, 2006). The existence of this glacial-type circulation has recently been found to be inseparable from an expanded
sea ice field (Ferrari et al., 2014; Sun and Matsumoto, 2010), which further restricts outgassing of carbon from nutrient-rich deep waters that upwell in the high latitudes. Other mechanisms, such as an equatorward shift in the westerly winds causing polar stratification (Toggweiler et al., 2006), greater brine rejection due to an expanded sea ice extent (Bouttes et al., 2010), and reduced interaction with bottom topography causing less diapycnal mixing (De Boer and Hogg, 2014), have also been implicated in the development of a glacial-type ocean circulation.

However, the most promising explanations of the decline in atmospheric $CO_2$ during glacial periods involve ocean biogeochemical changes in concert with reorganisations of the global overturning circulation (Hain et al., 2010; Sigman et al., 2010). Increased glacial productivity, first postulated by Broecker (1982) and now known to be driven by an increased deposition of aeolian dust to the Southern Ocean (Martinez-Garcia et al., 2014), is an established feature of the glacial sub-Antarctic
Southern Ocean. The Southern Ocean represents the most important region of carbon outgassing to the atmosphere because of the circumpolar extent of deep water upwelling (Burke and Robinson, 2012). Enhanced export production in this region is thus a prime candidate for explaining a large portion of the glacial-interglacial $CO_2$ difference. This has been demonstrated by models of varying complexity (Brovkin et al., 2012; Hain et al., 2010; Menviel et al., 2012).





In numerous other regions, however, productivity appears to have been reduced during glacial climates. The affected regions include waters south of the Antarctic Polar Front (Francois et al., 1997; Jaccard et al., 2013), the North Pacific (Crusius et al., 2004; Jaccard et al., 2005; Kohfeld and Chase, 2011; Ortiz et al., 2004), tropical Indian Ocean (Singh et al., 2011) and the Equatorial Pacific (Costa et al., 2016; Herguera, 2000; Loubere et al., 2007). A weaker export production in these regions would have offset a strengthened biological pump in the Sub-Antarctic, thereby weakening the ability of the ocean to store carbon at the LGM. Whether the strengthening of the biological pump in the glacial sub-Antarctic was able to outweigh losses in productivity in other regions requires further testing. This has led some authors to look for alternative biological mechanisms, notably temperature-dependent remineralisation (Chikamoto et al., 2012; Matsumoto, 2007; Menviel et al., 2012) and an altered $CaCO_3$:$C_{organic}$ export production ratio (Archer and Maier-Reimer, 1994; Lerman and Mackenzie, 2005; Sigman et al., 1998), to explain the net flux of carbon into the ocean.

Therefore, numerous physical and biogeochemical changes have been associated with a glacial ocean and all have been identified in some respect as important drivers of the carbon cycle. Now, recent insights into the distributions of dissolved oxygen (Jaccard et al., 2014) and carbonate species (Yu et al., 2014) within the glacial ocean provide new opportunities to identify which combination of physical and biogeochemical changes could have realistically sequestered carbon within the ocean at the LGM. Following the experiments conducted by Tagliabue et al. (2009), we use an Earth System Model with attached biogeochemistry, CSIRO Mk3L-COAL, to test current theories against these new insights. Using our simulated LGM ocean state, we provide a new perspective on the mechanisms responsible for the 80-100 ppm drawdown in atmospheric $CO_2$ during glacial cycles and demonstrate the importance of marine biogeochemistry to global climate.

## 2   Model and experiments

The model simulations were performed using the CSIRO Mk3L climate system model version 1.2 (Phipps et al., 2011, 2012), which includes components that describe the atmosphere, land, sea ice and ocean. The horizontal resolution of the atmosphere, land and sea ice models are $5.6° \times 3.2°$ in the longitudinal and latitudinal dimensions, respectively, with 18 vertical levels. The ocean model has a horizontal resolution of $2.8° \times 1.6°$ with 21 vertical levels. For this study, we conduct simulations using both the full climate system model and the stand-alone ocean model.

Two fully coupled model experiments were undertaken to simulate the Pre-Industrial (Cpl-PI) and Last Glacial Maximum (Cpl-LGM) climates. The Cpl-PI climate was obtained by forcing the model with an atmospheric $CO_2$ concentration of 280 ppm and by prescribing 1950 CE values for the orbital parameters. This experiment was integrated for a total of 10,000 years (Phipps et al., 2013). The Cpl-LGM simulation followed the protocol developed by Phase III of the Palaeoclimate Modelling Intercomparison Project (PMIP3), with the exception that no changes were made to terrestrial topography, oceanic bathymetry or the positions of the coastlines. The atmospheric $CO_2$ equivalent concentration was set to 167 ppm, providing a radiative forcing





equivalent to the specified reductions in the atmospheric concentrations of $CO_2$, $CH_4$ and $N_2O$ from 280 ppm/760 ppb/270 ppb for pre-industrial simulations to 185 ppm/350 ppb/200 ppb for LGM simulations. The orbital parameters were set to values for 21 ka BP. The Cpl-LGM simulation was initialised from the state of the Cpl-PI simulation at the end of model year 100. The model was then integrated for a total of 3,900 model years, until it had reached quasi-equilibrium. Over this integration the

ocean experienced a slow drift in whole-ocean salinity so that 0.5 psu was added, reflecting the coupling between a cooler atmosphere and the ocean.

With the Cpl-LGM climate state a suite of different ocean biogeochemical simulations were made with slightly different parameterisations to explore the effect on the carbon cycle (Table 1). These experiments utilised Mk3L-COAL (Carbon-Ocean-

Atmosphere-Land), an enhanced version of the Mk3L climate system model which includes biogeochemical modules embedded within the ocean, atmosphere and terrestrial models. For a description of the ocean biogeochemistry the reader is directed towards Appendix A of Matear and Lenton (2014) and the experiments of Duteil et al. (2012).

A total of 8 ocean-only simulations with on-line biogeochemistry were undertaken. All experiments were forced by key bound-

ary conditions (wind stresses, temperature, salinity, incident radiation, sea ice and atmospheric pressure at sea level), which were obtained as averages over the final 50 years of the fully coupled model experiments. The heat and freshwater fluxes into the ocean were determined by relaxing the SST and SSS towards the prescribed fields using a 20 day timescale. Experiments O-PI1 and O-LGM1 represent standard Pre-Industrial and Last Glacial Maximum simulations with atmospheric $CO_2$ concentrations at 280 and 185 ppm, respectively. Experiments O-PI2 and O-LGM2 were exactly the same as experiments O-PI1

and O-LGM1, except that atmospheric $CO_2$ concentrations were switched to investigate the effect of physical changes on the storage of carbon in the ocean. For these experiments, the biogeochemical model was unmodified. The remaining experiments, O-LGM3 to O-LGM6, represent glacial ocean-only runs in which the ocean biogeochemistry was altered. These alterations were as follows:

**O-LGM3.** The scaling factor ($S^O_{npp}$) was increased by a factor of 10 (Eq. (1)) to increase the export of Particulate Organic Carbon (POC) from the surface ocean, and therefore strengthen the biological carbon pump. Increasing POC export in the LGM ocean was motivated by an enhanced delivery of iron to the surface ocean via aeolian dust at the LGM (Delmonte et al., 2004; Kawahata et al., 2000; Lambert et al., 2012; Martin, 1990; Martínez-Garcia et al., 2009; Martinez-Garcia et al., 2014; Watson et al., 2000).

$$POC = S^O_{npp} * V_{max} * \min(\frac{[PO_4]}{[PO_4] + P_k}, F(I)) * 12 \tag{1}$$

**O-LGM4.** The POC remineralisation depth was increased by changing the power law exponent ($Rem_{pwl}$) in Equation (2) from -0.9 to -0.7, which replicated a bulk shift of POC from the upper to the deep ocean. The motivation for increasing the amount of POC that reaches deeper levels is the expectation that a cooler ocean would reduce the rate of bacterial remineralisation of POC in the upper ocean (Rivkin and Legendre, 2001; Matsumoto, 2007). This change increased the simulated POC that



reaches the 1,000 m depth level from 12.5 % to 20 %.

$$Remin(z) = \min(1.0, (\frac{z_{thick}}{100.0})^{Rem_{pwl}}) \qquad (2)$$

**O-LGM5.** Export production of Particulate Inorganic Carbon (PIC) was turned off by setting the rain ratio ($R_{PIC}$) of PIC:POC to zero in Equation (3). A reduction in PIC export at the LGM would increase the carbon content of the ocean, as marine calcifiers release $CO_2$ during formation of their shells and enhanced carbon loss via outgassing. The motivation for reducing PIC export in the glacial ocean is also temperature related, as there is a strong positive relationship between temperature and calcification (Lough and Barnes, 2000).

$$PIC = POC * R_{PIC} \qquad (3)$$

**O-LGM6.** All three modifications to ocean biogeochemistry were employed. All ocean-only simulations were integrated for 10,000 years to ensure that the ocean carbon cycle reached a steady state.

To assess whether the behaviour of the biogeochemical tracers within the coupled model differed from those in the ocean-only model, we ran the coupled model with online ocean biogeochemistry for a further 1,000 years using the steady-state biogeochemical fields from the ocean-only experiments. This assessment was made using both the PI and LGM climates. For key diagnostics, such as the meridional overturning circulation, ocean carbon content and global export production, the behaviour of the ocean-only simulation differed by less than 1 % from the coupled simulations. Given the computational speed of the ocean-only model, these experiments provide an ideal platform to test the sensitivity of the ocean biogeochemical fields to the parameterisations used in the biogeochemical model.

# 3 Results and discussion

In the following, we first discuss the simulated physical changes to the ocean observed between the Cpl-PI and Cpl-LGM simulations. Second, we discuss how the ocean biogeochemical fields differed between the O-PI1 and O-LGM1 simulations, which were forced with the output of the coupled simulations. Finally, we explore how modifying biogeochemical parameterisations alters the biogeochemistry, including changes to carbon storage, export production, aragonite saturation state and dissolved oxygen, that reconcile our simulated glacial ocean with what is considered realistic according to palaeoclimate proxy records.

## 3.1 LGM climate: physical fields

### 3.1.1 Sea surface temperature (SST)

The simulated change in SST between the Cpl-PI and the Cpl-LGM simulations shows a similar magnitude and spatial structure to proxy reconstructions and prior modelling studies, with greatest cooling in the equatorial oceans, high latitudes and



eastern boundary currents, and the least cooling in the subtropics and western boundary current regions (Fig. 1; Table 2). The global SST mean of the Cpl-LGM was 3.2 °C cooler than the Cpl-PI. This change falls within the range of estimates (∼2-4 °C) produced by other climate models (Alder and Hostetler, 2015; Annan and Hargreaves, 2013; Braconnot et al., 2007; Ganopolski et al., 1998; Kitoh et al., 2001; Shin et al., 2003; Smith and Gregory, 2012), but sits towards the cooler limits

of previous multiproxy SST reconstructions that estimate a change of 2 ± 1.8 °C (Ballantyne et al., 2005; McIntyre et al., 1976; Waelbroeck et al., 2009). However, a recent reanalysis of the proxy data presented by Waelbroeck et al. (2009) showed past estimates may have underestimated cooling by as much as 50 % (Ho and Laepple, 2015). This finding reconciles some disagreement between climate models and palaeoproxies, and places our simulated cooling of 3.2 °C well within the bounds of uncertainty in reconstructions.

Regionally, the greatest cooling took place in the high latitudes and in the Equatorial Pacific, where temperatures were in excess of 4 °C cooler than the Cpl-PI climate. Meanwhile, the Western Pacific Warm Pool, subtropical gyres and western boundary currents cooled less (0.5-3.0 °C). Again, proxy (Bostock et al., 2013; Gersonde et al., 2003, 2005; Kaiser et al., 2005; Kucera et al., 2005; Lamy et al., 2004; Lüer et al., 2009; Martínez-Garcia et al., 2009; Waelbroeck et al., 2009) and cli-

mate model simulations (Alder and Hostetler, 2015; Annan and Hargreaves, 2013, 2015; Shin et al., 2003) are consistent with both the magnitude and spatial pattern of cooling. Enhanced cooling in the high latitudes and in the eastern boundary currents generated strong zonal and meridional temperature gradients relative to Cpl-PI SST. There is a consistent regional pattern to SST cooling in the LGM emerging from proxy and model simulations (Annan and Hargreaves, 2013; Braconnot et al., 2007; Felis et al., 2014) that is broadly consistent with our simulated cooling.

Where there is still large uncertainty in SST change at the LGM is in the tropical ocean (see Annan and Hargreaves, 2015, for a review). The Cpl-LGM cooling of 3.3 °C across the tropical ocean (15° S - 15° N) is greater than other simulations (Annan and Hargreaves, 2013; Ballantyne et al., 2005; Braconnot et al., 2007; Ganopolski et al., 1998; Kitoh et al., 2001; Smith and Gregory, 2012), but falls well within the −5.1 to −2.17 °C estimated by Ho and Laepple (2015). Regionally, climate models

and proxies both agree that cooling in the tropical Atlantic Ocean probably exceeded cooling in the tropical Pacific and Indian Oceans by roughly 1 °C (Ballantyne et al., 2005; Ganopolski et al., 1998; Kitoh et al., 2001; Kucera et al., 2005; Otto-Bliesner et al., 2009; Waelbroeck et al., 2009). In contrast, the tropical Pacific Ocean cooled by 2 °C more than the Tropical Atlantic and Indian Oceans in the Cpl-LGM simulation. Although SSTs in the east equatorial Pacific have been reported as 1.5-3.0 °C cooler than the PI (Dubois et al., 2014; Kucera et al., 2005), the simulated cooling over much of the tropical Pacific appears

excessive compared to previous studies (Ballantyne et al., 2005; Braconnot et al., 2007; Chen et al., 2005).

### 3.1.2 Sea ice extent

While true estimates of sea ice coverage in the PI climate can only be inferred from whaling records, our Cpl-PI sea ice extents are consistent with estimates made using satellite measurements during the 1979-1987 period (Gloersen et al., 1993, Table 2).



These measurements represent the first global estimates of sea ice coverage, and although some evidence indicates that sea ice has declined by 20 % since the 1950's (Curran et al., 2003), the strong agreement between the Cpl-PI sea ice fields and the observations of Gloersen et al. (1993) provide a good benchmark for assessing LGM sea ice changes.

Associated with cooler SSTs, sea ice coverage (areas with sea ice concentration at or in excess of 15 %) was greatly expanded in the Cpl-LGM for both hemispheres relative to the Cpl-PI (Fig. 2, Table 2). In the Southern Hemisphere, total sea ice coverage increased by ∼120 % and ∼225 % at its seasonal maximum and minimum, respectively, relative to the Cpl-PI. In the Northern Hemisphere, total sea ice coverage increased by ∼145 % and ∼105 % at its seasonal maximum and minimum, respectively, relative to the Cpl-PI. These increases correspond to equatorward expansions of the sea ice field of between 5-10°
around the Southern Ocean, and in excess of 15° in both the North Atlantic and Pacific Oceans. The strong expansion of sea ice aligns with a recent theory postulated by Ferrari et al. (2014), who argued that the changes evident in global overturning circulation at the LGM (discussed in the next section), are inherently linked to an expanded sea ice field during both the winter and summer seasons.

The simulated expansion of sea ice around much of the Southern Ocean agrees well with proxy reconstructions. Maximum sea ice extent reached as far north as 47° S in both the Atlantic and Indian sectors and 57° S in the Pacific sector of the Southern Ocean (Gersonde et al., 2005, Fig. 2). This magnitude of growth in the Atlantic and Indian sectors has been tested and largely supported by a few subsequent studies (Collins et al., 2012; Xiao et al., 2016), and is consistent with our Cpl-LGM sea ice field. In the Pacific sector, however, the simulated maximum sea ice edge extends well equatorward of the 57° S suggested by
Gersonde et al. (2005) (Fig. 2). By comparing the coverage of sea ice in the Southern Hemisphere of the Cpl-LGM ($\sim46\times10^6$ km$^2$) with that estimated by Gersonde et al. (2005) ($\sim39\times10^6$ km$^2$), we can attribute the simulated excess of sea ice in the glacial Southern Ocean to a possible overestimate in the Pacific sector.

In the North Atlantic, perennial sea ice cover was present in the Greenland Sea and Fram Strait during the LGM (Müller
et al., 2009; Telesiński et al., 2014). There is also evidence that winter sea ice reached south of Iceland to fill much of the Labrador Sea (Pflaumann et al., 2003) and extended along the eastern Canadian margin (De Vernal et al., 2000, 2005). Meanwhile, the central and eastern parts of the subpolar North Atlantic are thought to have been largely ice-free (Pflaumann et al., 2003). In the North Pacific, proxy reconstructions suggest strong cover in the Okhotsk Sea (Sakamoto et al., 2005; Nürnberg and Tiedemann, 2004; Yamazaki et al., 2013), the Japan Sea (Ikehara, 2003) and the western Bering Sea (Ovsepyan et al., 2013;
Riethdorf et al., 2013b, a), with seasonally ice-free conditions in the central west (Jaccard et al., 2005). Sea ice reconstructions are lacking in the central and eastern North Pacific Ocean, but climate models that completed the PMIP3 LGM experiment simulate stronger sea ice presence in the western margins of the Northern Hemisphere basins (Fig. 2). The Cpl-LGM sea ice in this study is broadly consistent with the palaeo evidence in the North Atlantic, but an intense and year-round cover developed over the central North Pacific that contrasts directly with the findings of Jaccard et al. (2005), whom argued for ice-free conditions during the summer. Furthermore, the expansion of Cpl-LGM sea ice is greater than other PMIP3 climate models, which



places our simulated sea ice field towards the higher bounds predicted by climate system models for the LGM.

### 3.1.3 Meridional overturning circulation

The changes observed in the surface ocean within the Cpl-LGM climate were accompanied by changes in the global meridional overturning circulation (Fig. 3; Table 2). The rate of Antarctic Bottom Water (AABW) formation, defined here as the strongest negative cell south of 60° S, was greater in the Cpl-LGM than the Cpl-PI. AABW formation rates approached 25 Sv in the Cpl-LGM, about double that of the Cpl-PI. The greater formation rate increased the transport of AABW out of the Southern Ocean (northwards of 50° S) by roughly a factor of four from 17,440 to 71,664 Sv. This indicates that the increase in AABW

formation was also associated with an increase in the proportion of this water mass that was carried out of the Southern Ocean.

The formation rate of North Atlantic Deep Water (NADW), defined as the maximum cell of the North Atlantic streamfunction north of 45° N, was 12.5 Sv in the Cpl-LGM and 16 Sv in the Cpl-PI simulation, which equates to a ∼25 % reduction in NADW strength. Although this decrease was slight, the southward transport of NADW across the Equator was reduced three-fold from

18392 to 5391 Sv. Our simulated strength of NADW formation in the Cpl-LGM extends the lower bound of previous LGM simulations from 13.9 Sv to 12.5 Sv (Otto-Bliesner et al., 2007). The weakened formation of NADW was also associated with its shoaling from approximately 3,000 m in the PI to 1,500 m in the Cpl-LGM. Thus, in the Cpl-LGM the water mass below 1,500 m was characterised by AABW.

The changes to AABW and NADW circulation were conducive to the development of a global overturning circulation dominated by dense water from the Southern Ocean. An altered global overturning circulation is now a widely recognised component of glacial climate states across the Pleistocene (Broecker, 2013). In fact, several authors now attribute at least half of the glacial-interglacial atmospheric $CO_2$ difference to circulation changes in the ocean (Broecker et al., 2015; Kohfeld et al., 2005; Sigman et al., 2010). Theoretical, proxy and model-based research is now beginning to converge on the large-scale characteristics of

the glacial ocean circulation, where AABW was more dominant due to a combination of expanded sea ice, enhanced brine rejection causing denser bottom waters, and reduced diapycnal mixing (Adkins, 2013; De Boer and Hogg, 2014; Ferrari et al., 2014; Skinner et al., 2010; Watson and Naveira Garabato, 2006). The neutral density boundary between a northward flowing AABW and a southward flowing NADW was also shoaled substantially at the LGM in comparison with the current climate. Numerous palaeonutrient tracers support the presence of AABW within the deep North Atlantic Ocean at the LGM (Curry and

Oppo, 2005; Duplessy et al., 1988; Keigwin, 2004; Marchitto and Broecker, 2006; Oliver et al., 2010). The maximum depth of NADW flow was displaced to above 2,000 m as a direct result, and the shoaling of NADW facilitated the development of a saltier, more stratified glacial deep ocean (Adkins, 2013). Ferrari et al. (2014) have interpreted these changes as inextricably linked to the expansion of sea ice in the Southern Ocean, which caused a greater proportion of Circumpolar Deep Water to rise into a zone of negative buoyancy flux and thereby produce greater quantities of denser AABW. Imposing only the orbital





parameters and atmospheric radiative forcing of the LGM, our Cpl-LGM simulation was able to reproduce these features of a glacial ocean circulation.

## 3.2 LGM climate: biogeochemical fields

5 The physical changes in the ocean between the Cpl-PI and the Cpl-LGM, as described above, caused significant changes in ocean biogeochemistry within the ocean-only simulations (Table 3). To assist in the discussion of the large-scale biogeochemical changes we divide the upper and the deep ocean based on the 2,000 m depth. This approach also allows for more clearly distinguishing between changes to the global overturning circulation, air-sea exchange and biological processes on the biogeochemical fields.

### 3.2.1 Carbon

For the ocean-only experiment O-LGM1, the carbon content of the ocean was 517 Pg C less than the O-PI1 experiment (Fig. 4; Table 3). This change in carbon content reflects the combined effects of an altered ocean physics, which increased carbon in the deep ocean by 322 Pg C, and a lower atmospheric partial pressure, which caused 839 Pg C of carbon to be lost from the 15 upper ocean. The air-sea gas exchange scheme in CSIRO Mk3L-COAL is based on the parameterisation given by Wanninkhof (1992), and therefore regardless of the increase in solubility in the upper ocean that is caused by cooling, the decrease in partial pressure in the atmosphere ensured that the upper ocean DIC equilibrated at a lower concentration.

To quantify the carbon gain caused by a cooler ocean with expanded sea ice and an altered overturning circulation, the ocean-20 only experiment, O-LGM2, forced by the Cpl-LGM output was run with a PI atmospheric $CO_2$ concentration of 280 ppm. The ocean carbon content of O-LGM2 was increased by 1,127 Pg C relative to O-PI1. The gain of carbon in O-LGM2 confirmed that the glacial ocean was indeed conducive to partitioning greater quantities of carbon in the deep ocean, which can largely be attributed to the increased formation rate of AABW. AABW has recently been identified as eliciting the greatest response in atmospheric $CO_2$ of all major ocean water masses (Menviel et al., 2015). However, total carbon content decreased in ex-25 periment O-LGM1, which indicated that the the loss from the upper ocean due to equilibration with a lower atmospheric $CO_2$ concentration outweighed the gains caused by an altered physics. To quantify this effect, the ocean-only experiment, O-PI2, forced by the Cpl-PI output was run with a LGM atmospheric $CO_2$ concentration of 185 ppm. The ocean carbon content of O-PI2 was reduced by 1,486 Pg C relative to O-PI1, and this confirmed that changes in the carbon content of the ocean are highly dependent on atmospheric carbon. Although the contribution of physical changes to ocean carbon storage was significant at 30 1,127 Pg C, it could not in isolation outweigh the loss of 1,486 Pg C caused by equilibration with lower atmospheric $CO_2$.



### 3.2.2 Nutrients and export production

Phosphate ($PO_4$) concentrations in experiment O-LGM1 also declined in the upper ocean and increased in the deep ocean (Fig. 5). Like DIC, this reorganisation was driven by a strengthened AABW circulation cell and is consistent with proxy reconstructions. Cadmium and $\delta^{13}C$ measurements from the Atlantic Ocean show increased nutrient concentrations in the deep ocean, but reduced levels above 2,000 m at the LGM (Boyle, 1992; Gebbie, 2014; Marchitto and Broecker, 2006; Tagliabue et al., 2009).

A direct consequence of the redistribution of $PO_4$ was the reduction in the production of particulate organic matter across many regions of the O-LGM1 ocean (Fig. 5). With the exception of the South Pacific and isolated areas in the subtropics, export production in the O-LGM1 experiment decreased relative to the O-PI1 experiment, so that global export production was 56 % of O-PI1. The global reduction was also illustrated by a decrease in regenerated carbon ($DIC_{org}$), which indicates a weakened biological carbon pump (Table 3). The magnitude of the reduction in export production for the O-LGM1 experiment sits outside the range of 76-83 % estimated using oxygen isotopic measurements (Blunier et al., 2002) and in the opposite direction to the conclusions of Galbraith and Jaccard (2015), who argued for a net strengthening of the glacial biological pump at the LGM. The strong reduction of export production can be attributed, in part, to a large decrease in export production from the sub-Antarctic zone. This feature is in direct conflict with palaeoproductivity proxies in the Atlantic and Indian sectors of the sub-Antarctic Ocean (Anderson et al., 2002, 2014; Chase et al., 2001; Jaccard et al., 2013; Nürnberg et al., 1997), and some parts of the Pacific sector (Bradtmiller et al., 2009; Lamy et al., 2014). Outside of the Southern Ocean, the reduction in export production in the O-LGM1 experiment is consistent with palaeoproductivity evidence (see Introduction).

### 3.2.3 Carbonate chemistry

The loss of phosphate from the upper ocean and its increase at depth was mirrored by changes in alkalinity, so that the more alkaline signature of AABW, relative to NADW, dominated the deep ocean in experiment O-LGM1. The redistribution of alkalinity matches the redistribution of salinity, where salinity decreased (by 0.87 psu) in the surface ocean and increased (by 2.21 psu) in the deep ocean (Fig. 6).

The aragonite saturation state ($\Omega$) in surface waters of experiment O-LGM1 agrees with proxy reconstructions of coral reef extent at the LGM. Surface $\Omega$ between 40° S and 40° N in O-LGM1 ($\Omega = 3.8$) was slightly lower than that of O-PI1 ($\Omega = 4.0$), but increased in the high latitude oceans (Fig. 7). The globally averaged surface $\Omega$ of O-LGM1 was therefore only slightly different from that of O-PI1, at $\Omega = 3.3$ and $\Omega = 3.4$, respectively. Consequently, the simulated $\Omega = 3.25$ isoline, the value at present used to define the location of viable coral reef conditions (Hoegh-Guldberg et al., 2007), was nearly unchanged between the O-LGM1 and O-PI1 experiments. Recent sonar and coring in the southern portion of the Great Barrier Reef (Abbey et al., 2011; Yokoyama et al., 2011) have detected the presence of drowned coral reefs that existed at the LGM as far south as





reefs present today. Such observations are consistent with our O-LGM1 experiment and indicates that the extent of viable coral reefs was unlikely to have been significantly different at the LGM relative to today.

However, the magnitude of increase in alkalinity in the glacial deep ocean, which was not accompanied by a stoichiomet-

rically matched increase in DIC, caused unrealistic increases in the aragonite saturation horizon ($\Omega = 1$), otherwise known as the lysocline. The average position of the aragonite saturation horizon doubled in depth from 1526 m in O-PI1 to 2944 m in O-LGM1. Outside of the eastern tropical Pacific the entire water column of experiment O-LGM1 was super-saturated for aragonite (Fig. 8). There is good evidence that the mean position of the lysocline was not appreciably different at the LGM as compared to the Late Holocene (Yu et al., 2014). This information places the simulated lysocline ($\Omega = 1$) of O-LGM1 as

unrealistic.

### 3.2.4 Dissolved oxygen

Experiment O-PI1 produced a global average oxygen concentration of $\sim181$ mmol $O_2$ m$^{-3}$, similar to the PI global average of about 178 mmol $O_2$ m$^{-3}$ (Garcia, 2005). The combination of cooler SSTs, an enhanced subduction of AABW and the

reduction in export production in experiment O-LGM1 dramatically increased the oxygen levels in both the upper and deep ocean by $\sim80$ and $\sim120$ mmol m$^{-3}$, respectively, which constitutes a global increase of 55 % (Fig. 9; Table 3).

The increase in dissolved oxygen in O-LGM1 was considerable, but agreed well with proxy reconstructions for the upper ocean. The oxygen-poor intermediate waters of the western North Pacific (Ishizaki et al., 2009; Shibahara et al., 2007), eastern

North Pacific (Cannariato and Kennett, 1999; Cartapanis et al., 2011; Chang et al., 2014; Dean, 2007; Nameroff et al., 2004; Pride et al., 1999; Ohkushi et al., 2013; van Geen et al., 2003), eastern South Pacific (Martinez et al., 2006; Muratli et al., 2010; Salvatteci et al., 2016), Equatorial Pacific (Leduc et al., 2010) and Indian Ocean (Reichart et al., 1998; Suthhof et al., 2001; van der Weijden et al., 2006) were better oxygenated at the LGM relative to the PI climate. An important consequence of oxygenating the upper ocean is a reduction in the strength of denitrification in the these regions. Sedimentary $\delta^{15}$N records

suggest that global aggregate rates of water column denitrification rates over the past 200 kya were lower during glacial periods and higher during interglacial periods (Galbraith et al., 2004), and this is consistent with the simulated oxygenation of the upper ocean.

However, dissolved oxygen concentrations in the deep ocean increased to an average of 301 mmol $O_2$ m$^{-3}$ in O-LGM1,

and this contrasts starkly with existing palaeoclimate reconstructions. Deep waters of the Indian (Murgese et al., 2008; Sarkar et al., 1993; Schmiedl and Mackensen, 2006), North Atlantic (Hoogakker et al., 2014), Southern Ocean (Chase et al., 2001; Jaccard et al., 2016) and Equatorial Pacific (de la Fuente et al., 2015) were poorly ventilated at the LGM relative to the Holocene. Drawing from a global compilation of like studies, Jaccard and Galbraith (2012) and Jaccard et al. (2014) demonstrated that the deep ocean was largely deoxygenated relative to the Holocene on a global scale. While the increase in oxygen concentrations





in the upper ocean aligned with the direction of change inferred from proxies, the response in the deep ocean can be considered unrealistic.

## 3.3 Importance of ocean biogeochemistry for climate

The spatial pattern of export production, the depth of the aragonite saturation horizon and the dissolved oxygen field of experiment O-LGM1 are outstanding in their disagreement with proxy evidence. Notably, experiment O-LGM1 lost a substantial quantity of carbon from the upper ocean in excess of the additions to the deep ocean caused by an altered physical ocean state. Experiment O-LGM1 was therefore unable to explain the glacial-interglacial drawdown of atmospheric $CO_2$, despite the presence of a physical ocean state within realistic bounds, generated by the Cpl-LGM experiment. If we are to reconcile the biogeochemistry of the glacial ocean with that inferred from proxy evidence, we must consider altering ocean biogeochemistry.

### 3.3.1 Reconciling the carbon budget

Three plausible modifications to the ocean biogeochemistry (see methods) were considered: (1) increased POC export production, (2) increased depth of POC remineralisation, and (3) reduced PIC export. In the following we step through the changes to carbon content caused by each modification, and the reader is directed to Table 3 for reference.

(1) **Experiment O-LGM3.** Although the the scaling factor controlling the export production of organic matter was increased 10-fold, the actual increase in POC export production averaged over the global ocean was more modest at roughly 30 %. Because most of the ocean within the O-LGM1 experiment became phosphate limited as greater quantities of nutrients were redistributed into the deep ocean, the increase in export production in experiment O-LGM3 was only felt in those regions where $PO_4$ was not limiting. The sub-Antarctic zone of the Southern Ocean experienced the greatest increase in export production (∼250 %) due to the increase in the scaling factor, followed by a few small regions along the Chilean margin and in the Northwest Pacific (Fig. 7). These regional responses caused the global net export production rate to increase from 4.48 to 5.92 Pg C yr$^{-1}$. Although this rate of POC export production was still lower than the O-PI1 experiment of 8.01 Pg C yr $^{-1}$, this increased carbon content by 188 Pg C.

(2) **Experiment O-LGM4.** The shift of organic matter to depth was associated with a global reduction in POC export production of ∼1.2 Pg C yr$^{-1}$ as remineralisation released $PO_4$ and DIC further from the photic zone. Despite the reduction in the biological pump, the bulk transfer of POC to depth generated an increase in ocean carbon storage of 150 Pg C.

(3) **Experiment O-LGM5.** The elimination of PIC in the simulated glacial ocean increased the solubility of $CO_2$ in the surface ocean and enabled the ocean to store an additional 262 Pg C.





Independently, none of the above modifications were able to increase ocean carbon content relative to the O-PI1 experiment (O-LGM3: $-329$ Pg C; O-LGM4: $-367$ Pg C; O-LGM5: $-255$ Pg C). However, by employing all three biogeochemical modifications in one experiment (**Experiment O-LGM6**), the glacial ocean was able to store an additional 843 Pg C more than experiment O-LGM1 and 326 Pg C more than O-PI1. This magnitude of increase is within the plausible bounds required to offset the loss of atmospheric and terrestrial carbon reported by Ciais et al. (2011) of $\sim 520 \pm 400$ Pg C at the LGM (Table 4).

### 3.3.2 Reconciling export production

Experiment O-LGM1 generated a strongly reduced POC export production across almost all regions of the global ocean (Fig. 5). Of the three biogeochemical modifications applied to the LGM ocean, only two had any effect on POC export, as the amount of PIC exported from the photic zone has no influence on the amount of POC export. Deepening the remineralisation of POC (O-LGM4) shifted a greater fraction of regenerated $PO_4$ into the deep ocean, which resulted in a global reduction of export production. Increasing the scaling factor (O-LGM3), however, caused an increase in global export production from 4.48 to 5.92 Pg C yr$^{-1}$. Most of this increase occurred in the Southern Ocean, particularly the sub-Antarctic zone, and in a few isolated pockets in the Northwest Pacific and North Atlantic (Fig. 10).

The increase in the scaling factor dominated the change in export production produced when combining all three biogeochemical modifications (O-LGM6). The strong increase in export production observed in the sub-Antarctic was clearly replicated within this experiment and reconciles our simulated export production field with current evidence of productivity at the LGM. In the Southern Ocean, the Atlantic and Indian sectors of the sub-Antarctic zone experienced a greater flux of organics to the sediments (Anderson et al., 2002, 2014; Chase et al., 2001; Jaccard et al., 2013; Nürnberg et al., 1997). Whether this was also the case for the Pacific sector remains under debate, with some evidence for increase (Bradtmiller et al., 2009; Lamy et al., 2014) conflicting with evidence for no change (Bostock et al., 2013; Chase et al., 2003). Meanwhile, it is widely accepted that waters south of the Antarctic Polar Front were reduced in their productivity (Bostock et al., 2013; Chase et al., 2003; Elderfield and Rickaby, 2000; Francois et al., 1997; Frank et al., 2000; Kohfeld et al., 2005; Kumar et al., 1995; Mortlock et al., 1991; Ninnemann and Charles, 1997; Shemesh et al., 1993), likely due to increased sea ice extent (Gersonde et al., 2003; Jaccard et al., 2013) and stratification (Anderson et al., 2014; Jaccard et al., 2005).

In experiment O-LGM6, net export production remained weakened by 3.19 Pg C yr$^{-1}$ relative to O-PI1 despite the application of biogeochemical modifications. This result is contrary to arguments for a strengthened biological pump at the LGM (Galbraith and Jaccard, 2015), and aligns more with the findings of Blunier et al. (2002). The net decline in export production observed in this study was dominated by the decline in tropical and subtropical waters. Many palaeoproductivity studies located outside of the sub-Antarctic zone have found weakened productivity at the LGM (Chang et al., 2014, 2015; Costa et al., 2016; Crusius et al., 2004; Kohfeld et al., 2005; Kohfeld and Chase, 2011; Jaccard et al., 2005; McKay et al., 2015; Ortiz et al., 2004;





Riethdorf et al., 2013b; Salvatteci et al., 2016; Singh et al., 2011; Thomas et al., 1995). Additionally, an enhanced utilisation of available nutrients in the sub-Antarctic zone (Martinez-Garcia et al., 2014) would reduce nutrient content of intermediate waters formed in the Southern Ocean and would thus reduce the delivery of nutrients to lower latitudes (Sarmiento et al., 2004). This mechanism coupled with cooler temperatures caused reductions in export production across much of the mid and lower

latitude oceans in experiment O-LGM6, which maintains the qualitative agreement between simulated and proxy observations (see Introduction). Hence, the good spatial agreement between O-LGM6 and palaeoproductivity proxies at the LGM gives confidence that a stronger biological pump in sub-Antarctic waters, combined with an expanded sea ice cover that limited air-sea exchange in the Antarctic Zone, was a key component for transferring carbon from the atmosphere to the ocean at the LGM.

### 3.3.3   Reconciling carbonate chemistry

There is good evidence that the mean position of the lysocline was not appreciably different at the LGM as compared to the Late Holocene (Yu et al., 2014). Because much of the ocean was saturated for aragonite in the O-LGM6 experiment, additional processes are required to shoal the aragonite saturation horizon ($\Omega = 1$) and thereby reconcile proxy evidence.

One mechanism to shoal the aragonite saturation horizon would be to reduce continental inputs of alkalinity at the LGM. However, the presence of glaciers, drier atmospheric conditions and the exposure of continental shelves due to lower sea level would have increased the supply of carbonates to the ocean (Gibbs and Kump, 1994; Riebe et al., 2004), thereby increasing ocean alkalinity and further deepening the aragonite saturation horizon. This mechanism has been largely refuted as having a significant effect on the glacial-interglacial difference in the carbon budget (Brovkin et al., 2007; Foster and Vance, 2006;

Jones et al., 2002), and can therefore be ignored.

The individual biogeochemical modifications were also insufficient to effectively shoal the depth of aragonite saturation to be consistent with palaeo evidence. However, combining all three modifications in experiment O-LGM6 shoaled the aragonite saturation horizon significantly (Fig. 11), with a globally-averaged position of 1818 m. Regionally, the aragonite saturation

horizon in the Pacific Ocean was deeper in experiment O-LGM6 relative to O-PI1, but was shallower in the Atlantic Ocean and within the Atlantic and Indian sectors of the sub-Antarctic zone. Remarkably, these positions relative to the PI climate are consistent with palaeoproxy reconstructions. A deepening of less than 1,000 m has been suggested in the North Pacific and Southern Ocean at the LGM (Anderson et al., 2002; Catubig et al., 1998), while other proxy evidence suggests that the lysocline in the Atlantic Ocean was shallower than the PI climate (Anderson et al., 2002).

However, an important caveat of this study is the exclusion of calcium carbonate ($CaCO_3$) burial within ocean sediments. Because this process is not included in the model, it is highly likely that the deepening of the aragonite saturation horizon that occurred in experiment O-LGM1 was too extreme. $CaCO_3$ burial lowers the alkalinity of the glacial ocean, and is therefore a negative feedback mechanism to changes in the position of the aragonite saturation horizon (see Sigman et al., 2010, for a



review). If the aragonite saturation horizon deepens, as found in O-LGM1, the burial of $CaCO_3$ would increase and cause a subsequent reduction in ocean alkalinity, the depth of the lysocline and the ability of the ocean to store carbon. By not taking this process into account in experiment O-LGM1, both the deepening of the aragonite saturation horizon and the atmospheric drawdown of carbon were overestimated.

However, the same reasoning can be applied to the experiments with biogeochemical modifications. If the burial of $CaCO_3$ was included in these experiments, the shoaling in the aragonite saturation horizon would have been somewhat mitigated by decreased $CaCO_3$ burial that increased ocean alkalinity. Consequently, the shoaling that was observed in these experiments was likely exaggerated, just as the deepening observed in experiment O-LGM1 was exaggerated. Again, this can be applied to changes in the carbon content of the ocean, as a shallower aragonite saturation horizon would have increased whole-ocean alkalinity and thereby increased the drawdown of atmospheric $CO_2$. This effect would have been particularly important for experiment O-LGM5, where inorganic carbon export was eliminated. If whole-ocean alkalinity was able to respond to the decrease in $CaCO_3$ rain, this would further increase the associated $CO_2$ drawdown. Therefore, the exclusion of $CaCO_3$ burial in experiment O-LGM6 caused an exaggerated shoaling of the aragonite saturation horizon and an underestimated increase in carbon content.

### 3.3.4 Reconciling dissolved oxygen

As discussed previously, the increase in oxygen concentrations of the upper ocean in experiment O-LGM1 is consistent with proxy evidence. All experiments with modified biogeochemistry, including O-LGM6, had little effect on the upper ocean oxygen concentration (Fig. 12; Table 3). Modifying the biogeochemistry did not compromise the good agreement between simulated and proxy reconstructions of oxygen concentrations in the upper ocean.

Modifying ocean biogeochemistry did, however, have a large effect on the oxygen concentrations of the deep ocean (Fig. 13). Increasing export production (O-LGM3) and deepening the remineralisation depth (O-LGM4) both reduced oxygen concentrations by 28 and 13 mmol m$^{-3}$, respectively. The combination of these modifications (O-LGM6) amplified their individual effects, so that deep ocean oxygen was reduced by 63 mmol m$^{-3}$ relative to the O-LGM1. The increased sensitivity of deep ocean oxygen to the combination of increased export production and a deeper remineralisation depth was due to an increase in the quantity of regenerated nutrients ($DIC_{org}$ and $P_{org}$) that resulted (Table 3). A greater proportion of regenerated nutrients relative to preformed nutrients at the LGM has been identified as a key driver of interior ocean deoxygenation (Jaccard and Galbraith, 2012; Sigman et al., 2010), and this process was captured in experiment O-LGM6.

While the combination of biogeochemical modifications (O-LGM6) did reduce deep ocean oxygen towards those concentrations equivalent to or lower than those of experiment O-PI1 in a number of areas (Fig. 13), by no means were average deep ocean concentrations (238 mmol m$^{-3}$) close to those of O-PI1 (181 mmol m$^{-3}$). The over-oxygenation of the deep ocean rel-



ative to proxy records may be resolved by (1) reducing the strength of the AABW circulation cell, and/or (2) further increasing export production, and/or (3) altering the spatial pattern of export production. These possibilities strongly indicate that global export production at the LGM may be underestimated by our simulations. Further investigation is required to reconcile our experiments with palaeo evidence of deep ocean deoxygenation.

## 4   Conclusions

In this study we have shown that physical changes to the ocean state, including an expanded sea ice field and an altered circulation, are not sufficient to explain the drawdown of 80-100 ppm $CO_2$ in the atmosphere of the LGM. While we demonstrate that

the physical ocean state at the LGM is indeed highly conducive to storing carbon, owing largely to an increased subduction of Antarctic Bottom Water, the tendency for the upper ocean to lose carbon through its equilibration with a lower atmospheric $CO_2$ concentration outweighs these gains. Thus, various biogeochemical modifications are necessary to overcome these losses and produce net gains of carbon in the ocean. The marine biogeochemical changes explored in this study were (1) an increase in export production consistent with greater iron fertilisation, (2) a shift of remineralisation to the deep ocean consistent with

cooler temperatures, and (3) a decrease in the production of Particulate Inorganic Carbon consistent with cooler temperatures. Only when all three changes were applied to a glacial ocean does the ocean carbon content increase sufficiently to account for the combined loss of carbon from the atmosphere and land at the LGM. Furthermore, their addition helps to reconcile unrealistic fields of export production, aragonite saturation state and dissolved oxygen produced by a simulation of the LGM without biogeochemical changes.

A key limitation of this study was the inability of biogeochemical modifications to truly deoxygenate the deep ocean on a global scale, as shown in palaeoclimate reconstructions. Either one or a combination of the following could resolve this inconsistency in our simulations: (1) the strength of the AABW circulation cell was too strong; (2) global export production was too weak; (3) the true spatial pattern of export production was not captured. Importantly, points (2) and (3) further implicate

ocean biogeochemical processes as a strong influence on climate. New focus should be applied to investigate these possibilities, including the use of multiple representations of the LGM physical ocean state and its effect on ocean biogeochemistry. Future work should also aim to include sedimentary processes, including carbonate burial, considering the importance of sediment processes to ocean biogeochemistry and climate.

*Acknowledgements.* Funding for this work was provided by the Australian Climate Change Science Program and CSIRO Wealth from

Ocean Flagship. An award under the Merit Allocation Scheme on the NCI National Facility at the Australian National University ensured that numerical simulations could be undertaken. This research was also supported under the Australian Research Council's Special Research Initiative for the Antarctic Gateway Partnership (Project ID SR140300001). The authors wish to acknowledge the use of the Ferret pro-





gram (http://ferret.pmel.noaa.gov/Ferret/) for the analysis undertaken in this work. The matplotlib package (Hunter, 2007), Iris and Cartopy

packages (http://scitools.org.uk/), and cmocean package (Thyng et al., 2016) were all used for producing visualisations.



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





**Table 1.** Summary of modelling experiments performed. An O before a model name denotes that it was an ocean-only simulation.

| Experiment | Model | Greenhouse Gas Forcing $(CO_2e)^a$ | Orbital Parameters | Comment |
|---|---|---|---|---|
| Cpl-PI | Coupled | 280 | 0 ka BP | Unmodified BGC |
| Cpl-LGM | Coupled | 167 | 21 ka BP | Unmodified BGC |

| Experiment | Model | Atmospheric $CO_2$ (ppm) | Climate State | Comment |
|---|---|---|---|---|
| O-PI1 | Ocean | 280 | PI | Unmodified BGC |
| O-PI2 | Ocean | 185 | PI | Unmodified BGC |
| O-LGM1 | Ocean | 185 | LGM | Unmodified BGC |
| O-LGM2 | Ocean | 280 | LGM | Unmodified BGC |
| O-LGM3 | Ocean | 185 | LGM | 10× POC export scaling increase |
| O-LGM4 | Ocean | 185 | LGM | Increased depth of POC remineralization[b] |
| O-LGM5 | Ocean | 185 | LGM | No PIC export |
| O-LGM6 | Ocean | 185 | LGM | BGC modifications of O-LGM3, O-LGM4 and O-LGM5 |

[a] Carbon Dioxide equivalents, corresponding to $CO_2$, $CH_4$ and $N_2O$ from 280 ppm/760 ppb/270 ppb for PI simulations to 185 ppm/350 ppb/200 ppb for LGM simulations.

[b] Power law exponent for POC remineralization changed from $-0.9$ to $-0.7$.



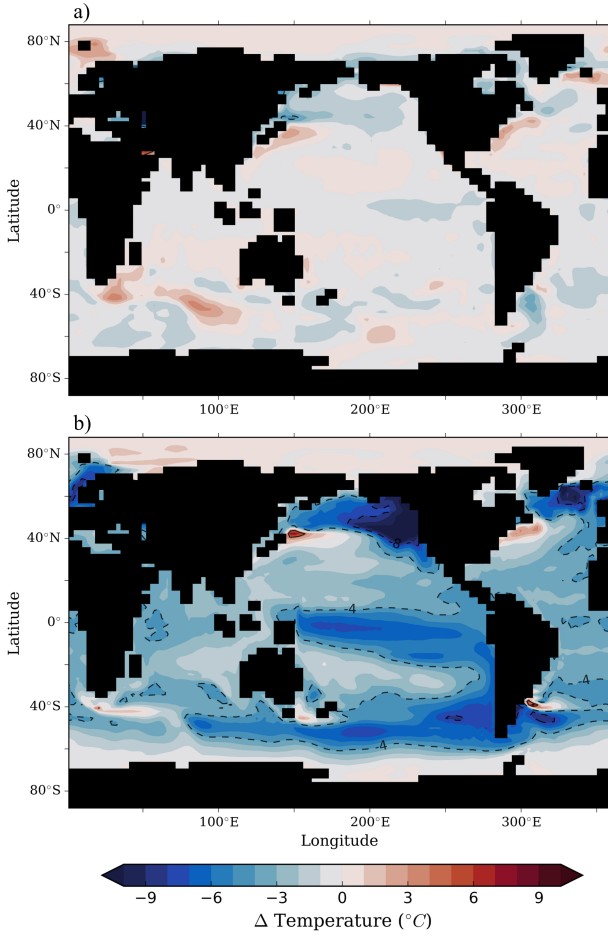

**Figure 1.** Annual sea surface temperature (SST) difference between (a) the coupled PI experiment, Cpl-PI, and the observations from Levitus (2001), and (b) the difference between the coupled LGM and PI experiments (Cpl-LGM −Cpl-PI). Solid contour lines denotes positive changes in SST by 4 and 8 °C, while negative changes in SST are denoted by dashed lines at 4 and 8 °C.



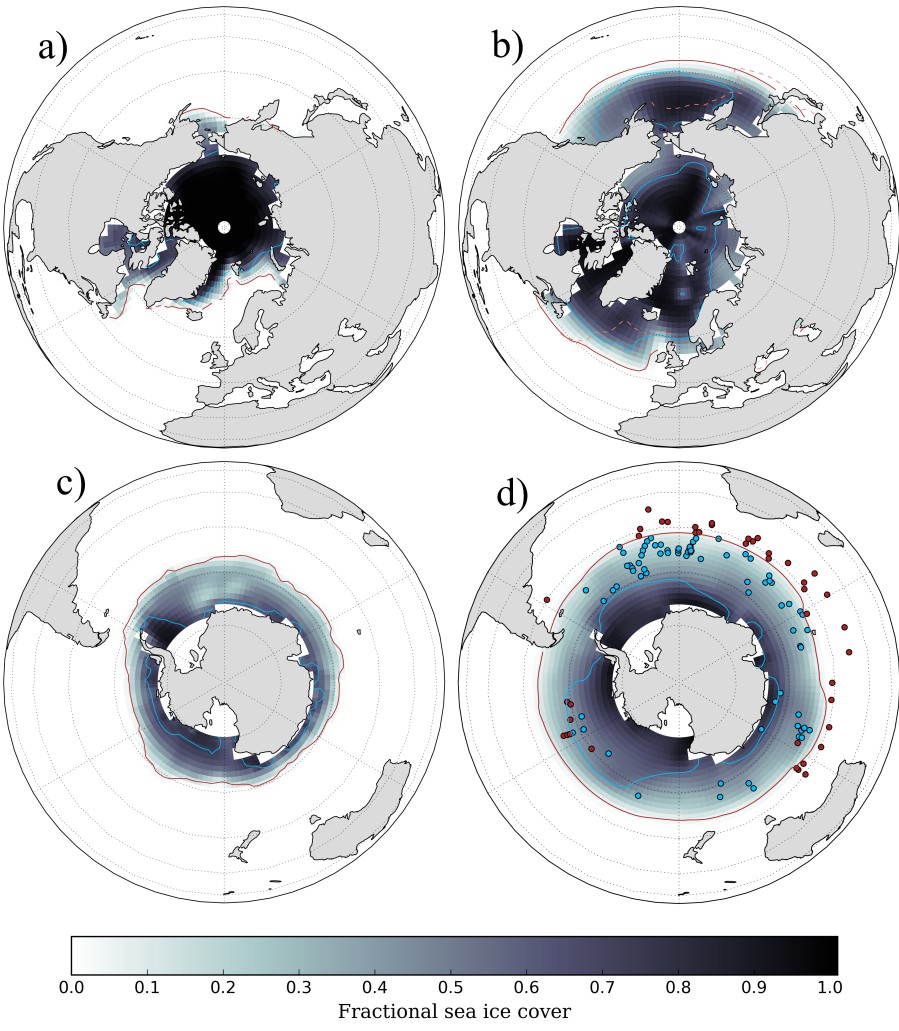

**Figure 2.** Annual average sea-ice cover for (a) the Cpl-PI Northern Hemisphere, (b) the Cpl-LGM Northern Hemisphere, (c) the Cpl-PI Southern Hemisphere, (d) the Cpl-LGM Southern Hemisphere. The red and blue contour lines in each projection represent the maximum and minimum seasonal sea ice extents (where sea ice concentration equals 15 % as per Gersonde et al. (2005)). In panel (b), the dashed orange contour line represents the maximum seasonal sea ice extent produced by the IPSL climate system model, which took part in the PMIP3 LGM experiment, and is broadly consistent with the results of other PMIP3 models. In panel (d), the coloured markers represent locations were winter sea ice was deemed to have been present (blue) and absent (red) at the LGM according to Gersonde et al. (2005).



**Table 2.** Changes in sea surface temperature (SST), sea ice extent and large scale circulation between the LGM and PI simulations. SST changes are compared with proxy reconstructions of SST generated by Waelbroeck et al. (2009) and Ho and Laepple (2015) whom use different proxies for their reconstructions to produce the differences depicted here and discussed in the text. The range of formation rates of Antarctic Bottom Water (AABW) and North Atlantic Deep Water (NADW) depicted here were calculated by Otto-Bliesner et al. (2007) from the CCSM and MIROC coupled model simulations of the LGM as part of the Palaeoclimate Modelling Intercomparison Project Phase II.

| | | $\Delta$ SST (°C) | | | |
|---|---|---|---|---|---|
| | | 15° S–15° N | 30° S–30° N | 60° S–60° N | 90° S–90° N |
| Simulated | Global | −3.3 | −3.2 | −3.9 | −3.2 |
| | Atlantic | −3.6 | −3.4 | −3.9 | −3.3 |
| Waelbroeck et al. (2009) | Global | −1.7 ± 1.0 | −1.5 ± 1.2 | −1.9 ± 1.7 | −1.9 ± 1.8 |
| | Atlantic | −2.9 ± 1.3 | −2.3 ± 1.5 | −2.6 ± 2.0 | −2.4 ± 2.2 |
| Ho and Laepple (2015) | Global | | −5.1 to −2.17 ± 1.43 | | −12.52 to −2.17 ± 1.43 |

| | | Sea ice extent ($10^6$ km$^2$) | | | |
|---|---|---|---|---|---|
| | | SH maximum | SH minimum | NH maximum | NH minimum |
| Simulated | PI | 20.8 | 4.12 | 15.0 | 9.73 |
| | LGM | 46.3 | 13.3 | 36.6 | 19.9 |
| (Gloersen et al., 1993) | PI | 19 | 3.5 | 16 | 9 |
| (Gersonde et al., 2005) | LGM | 39 | | | |

| | | Overturning metrics | |
|---|---|---|---|
| | | Max formation rate[a] (Sv) | Transport[b] (Sv) |
| Simulated | AABW (PI ‖ LGM) | 13.2 ‖ 25.1 | 17,440 ‖ 71,664 |
| | NADW (PI ‖ LGM) | 16.2 ‖ 12.5 | 18,392 ‖ 5,391 |
| PMIP2 models | AABW (PI ‖ LGM) | 10.4–16.2 ‖ 19.6–40.0 | |
| (Otto-Bliesner et al., 2007) | NADW (PI ‖ LGM) | 18.6–19.4 ‖ 13.9–30.7 | |

[a] Maximum formation rate of AABW is calculated as the most negative rate of overturning (Sv) in the Southern Ocean south of 60° S and deeper than 500 m. Maximum formation rate of NADW is calculated as the most positive rate of overturning (Sv) in the North Atlantic north of 45° N and deeper than 500 m.

[b] Yearly transport of AABW out of the Southern Ocean is calculated by integrating all negative values of the global meridional overturning streamfunction at 50° S across depth and time. Yearly transport of NADW out of the North Atlantic is calculated by integrating all positive values of the Atlantic meridional overturning streamfunction at 0° latitude across depth and time.



**Table 3.** Global ocean averaged diagnostics from the model simulations described in Table 1. The subscript organic refers to the inventory due to the remineralization computed from the apparent oxygen utilization. POC and PIC refer to the annual export of particulate organic and inorganic carbon from the upper 50 m, respectively. The tracer columns refer to global ocean inventory or global ocean mean values. Global inventory of phosphate was 2.68 Pmol in all simulations. White space for experiments O-PI2 and O-LGM2 indicate no difference with experiments O-PI1 and O-LGM1, respectively.

| Model | Atmospheric[a] $CO_2$ (ppm) | POC ($Pg\,C\,y^{-1}$) Global | POC Southern Ocean[d] | PIC ($Pg\,C\,y^{-1}$) Global | $\Delta DIC$[b] ($Pg\,C$) Global | Upper[e] | Deep[f] | $DIC_{org}$ ($Pg\,C$) | Oxygen (mean) ($mmol\,m^{-3}$) Global | Upper[e] | Deep[f] | Depth[c] (m) where $\Omega = 1$ |
|---|---|---|---|---|---|---|---|---|---|---|---|---|
| O-PI1 | 280 | 8.01 | 1.60 | 0.64 | 0 | 0 | 0 | 1650 | 181 | 182 | 180 | 1526 |
| O-PI2 | 185 | | | | −1486 | −801 | −686 | | | | | |
| O-LGM1 | 185 | 4.48 | 0.76 | 0.36 | −517 | −839 | 322 | 667 | 281 | 263 | 301 | 2944 |
| O-LGM2 | 280 | | | | 1127 | −40 | 1165 | | | | | |
| O-LGM3 | 185 | 5.92 | 4.06 | 0.47 | −329 | −933 | 603 | 772 | 272 | 271 | 273 | 2995 |
| O-LGM4 | 185 | 3.25 | 0.74 | 0.26 | −367 | −760 | 391 | 721 | 276 | 266 | 288 | 2994 |
| O-LGM5 | 185 | 4.48 | 0.76 | 0.00 | −255 | −447 | 190 | 667 | 281 | 263 | 301 | 2883 |
| O-LGM6 | 185 | 4.82 | 3.44 | 0.00 | 326 | −406 | 732 | 1004 | 252 | 265 | 238 | 1818 |

[a] Atmospheric $CO_2$ is prescribed in each experiment.

[b] Change in the ocean inventory of carbon relative to experiment O-PI1 with atmospheric $CO_2$ at 280 ppm.

[c] Where the aragonite saturation horizon exceeds the depth of the ocean, the deepest grid box was included in the averaging process.

[d] All grid boxes south of 45° S.

[e] All grid boxes above 2,000 m depth.

[f] All grid boxes below 2,000 m depth.





**Table 4.** The change in the LGM ocean carbon content relative to the PI ocean.

| Estimate | ΔCarbon |
| --- | --- |
| | (Pg C) |
| Ciais et al. (2011) | $520 \pm 400^{a}$ |
| PI - unmodified BGC[b] | $-520 \pm 400$ |
| PI - unmodified BGC ($CO_2$ of 185 ppm) | $-2006 \pm 400$ |
| LGM - unmodified BGC | $-1037 \pm 400$ |
| LGM - unmodified BGC ($CO_2$ of 280 ppm) | $607 \pm 400$ |
| LGM - modified BGC (O-LGM6)[c] | $-194 \pm 400$ |

[a] Estimate of increase in ocean carbon content during the LGM made by Ciais et al. (2011), whereby atmospheric carbon was reduced by $194 \pm 2$ Pg C and terrestrial carbon was reduced by $330 \pm 400$ Pg C.

[b] BGC refers to biogeochemistry.

[c] Assumes all three biological modifications that were postulated (see Table 1, experiments O-LGM3, O-LGM4 and O-LGM5) occurred to provide an upper bound estimate of ocean carbon storage.





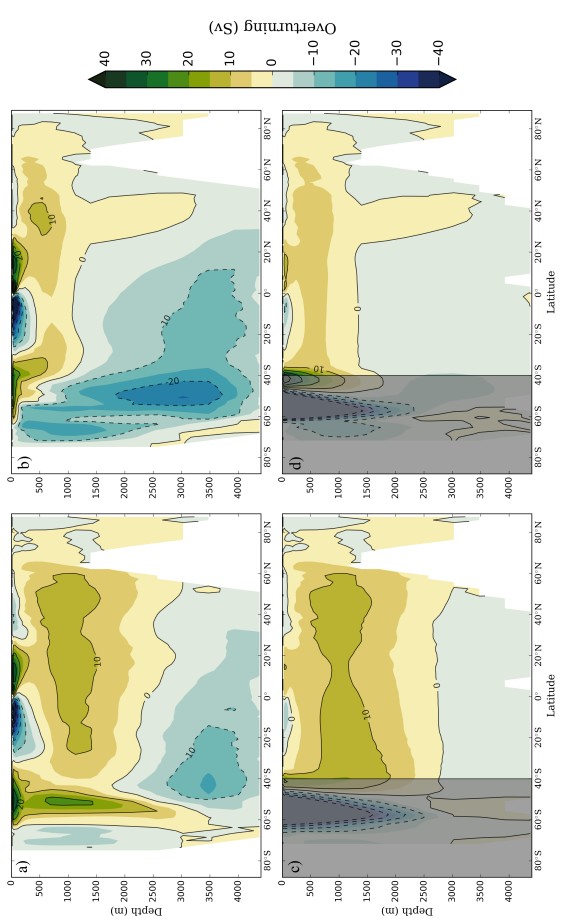

**Figure 3.** The upper panels depict the total meridional overturning streamfunction (Sv) for the global ocean in the (a) Cpl-PI and (b) Cpl-LGM simulations. The bottom panels depict the total meridional overturning streamfunction (sv) for the Atlantic ocean in the (c) Cpl-PI and (d) Cpl-LGM simulations. Note that those latitudes corresponding to the Southern Ocean are obscured for panels (c) and (d) in the Atlantic Ocean, as these overturning velocities are invalid considering that waters can exit to the east and west and that the streamfunction does not account for these losses.





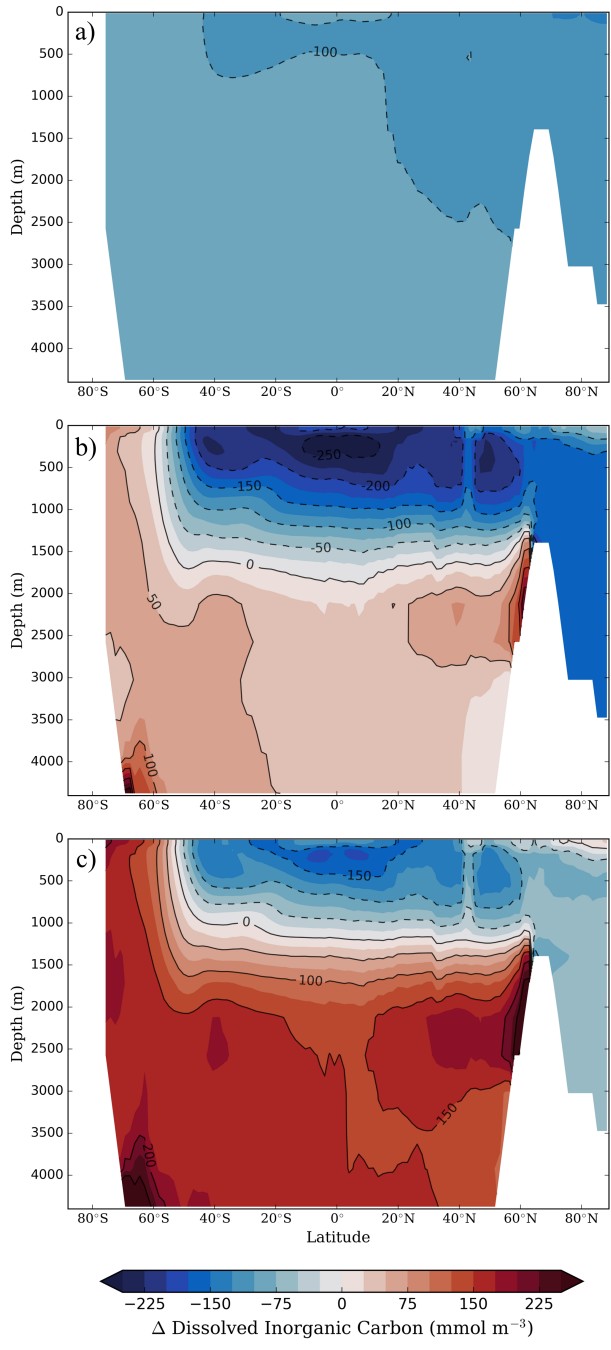

**Figure 4.** The zonally-averaged change in the concentration of dissolved inorganic carbon (mmol m$^{-3}$) relative to the O-PI1 experiment with an atmospheric $CO_2$ concentration of 280 ppm for (a) the PI ocean with an atmospheric $CO_2$ concentration of 185 ppm (O-PI2), (b) the LGM ocean with an atmospheric $CO_2$ concentration of 185 ppm (O-LGM1), and (c) the LGM ocean with an atmospheric $CO_2$ concentration of 280 ppm (O-LGM2).



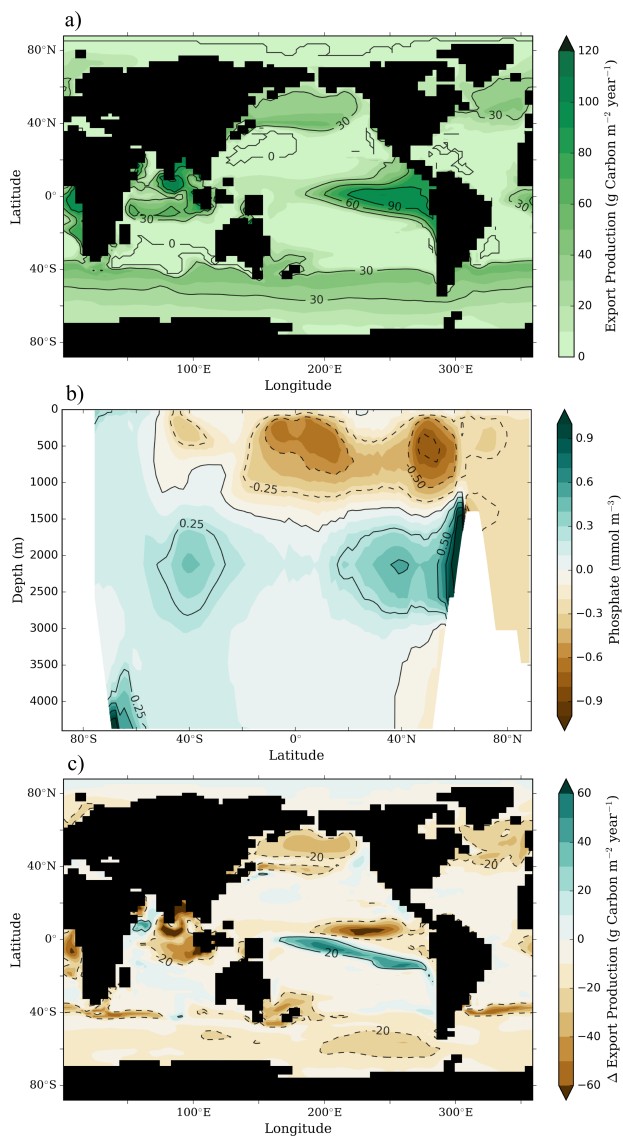

**Figure 5.** Changes in the export production of Particulate Organic Matter (POC) and Phosphate concentrations between the O-LGM1 and O-PI1 experiments. (a) Annually averaged export of POC from the upper 50 m (g Carbon m$^{-2}$ year$^{-1}$) for O-PI1, (b) the O-LGM1 − O-PI1 difference in Phosphate concentrations (mmol m$^{-3}$), and (c) the O-LGM1 − O-PI1 difference in export production of POC from the upper 50 m (g Carbon m$^{-2}$ year$^{-1}$).



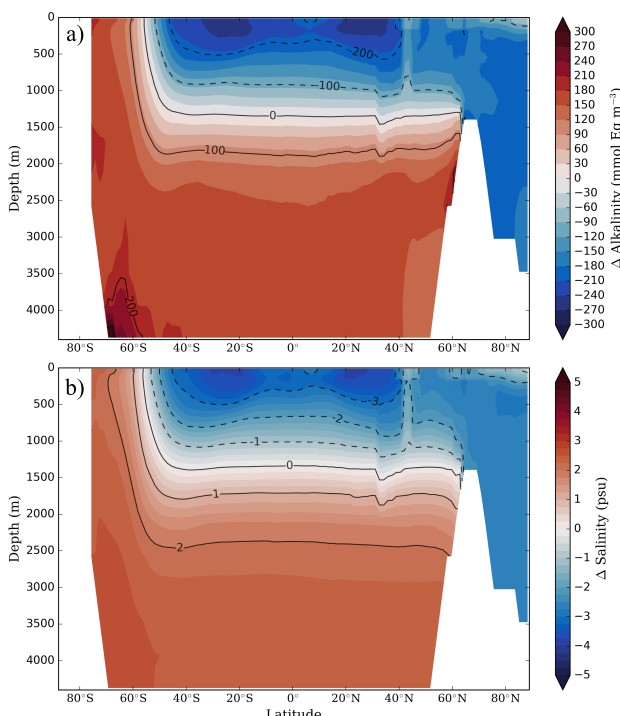

**Figure 6.** Change in the zonally-averaged global distribution of (a) alkalinity (mmol Eq m$^{-3}$), and (b) salinity (psu) between the O-LGM1 and O-PI1 experiments (O-LGM1 − O-PI1). Despite the strong reduction in salinity in upper ocean of the O-LGM1 experiment relative to O-PI1, the whole-ocean salt content increased by 0.5 psu.





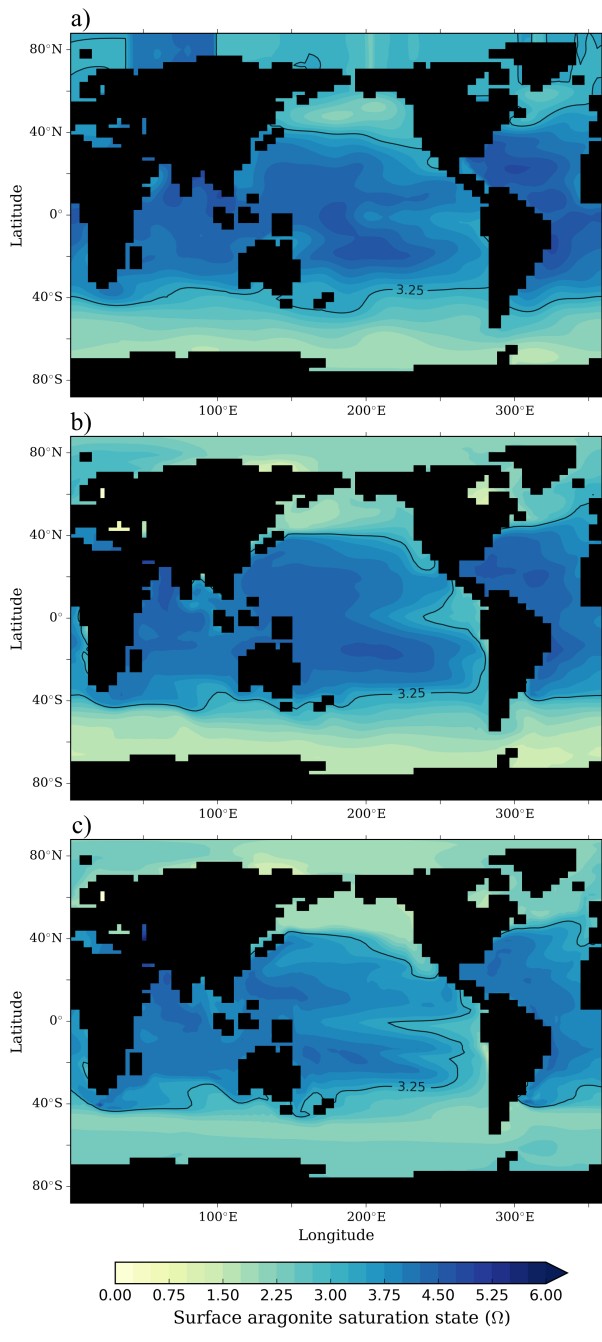

**Figure 7.** The annual average surface aragonite saturation state (Ω) calculated from (a) the observations of Key et al. (2004), (b) the O-PI1 experiment, and (c) the O-LGM1 experiment.





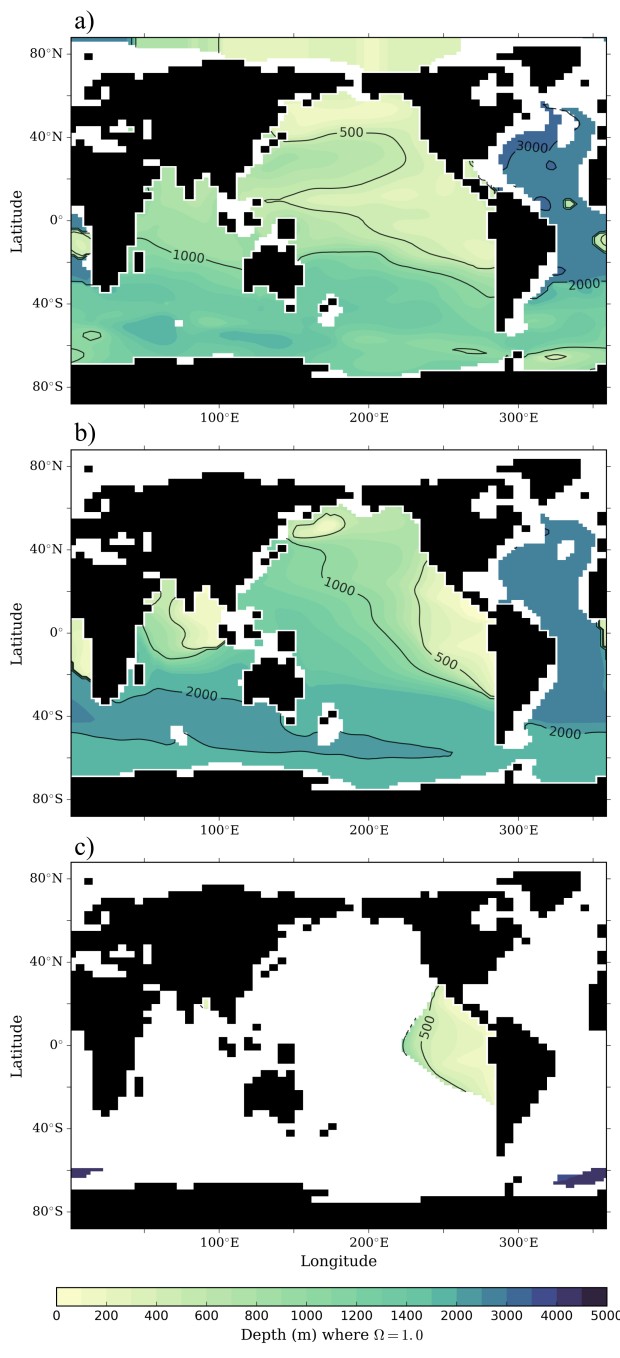

**Figure 8.** The depth of the aragonite saturation horizon ($\Omega = 1$) calculated from (a) the observations of Key et al. (2004), (b) the O-PI1 experiment, and (c) the O-LGM1 experiment. The contour lines represent 500, 1000, 2000 and 3000 m depth. Note that the O-LGM1 experiment, which is unmodified in its biogeochemistry relative to the O-PI1 experiment, is completely saturated in aragonite across the majority of the ocean (white space).





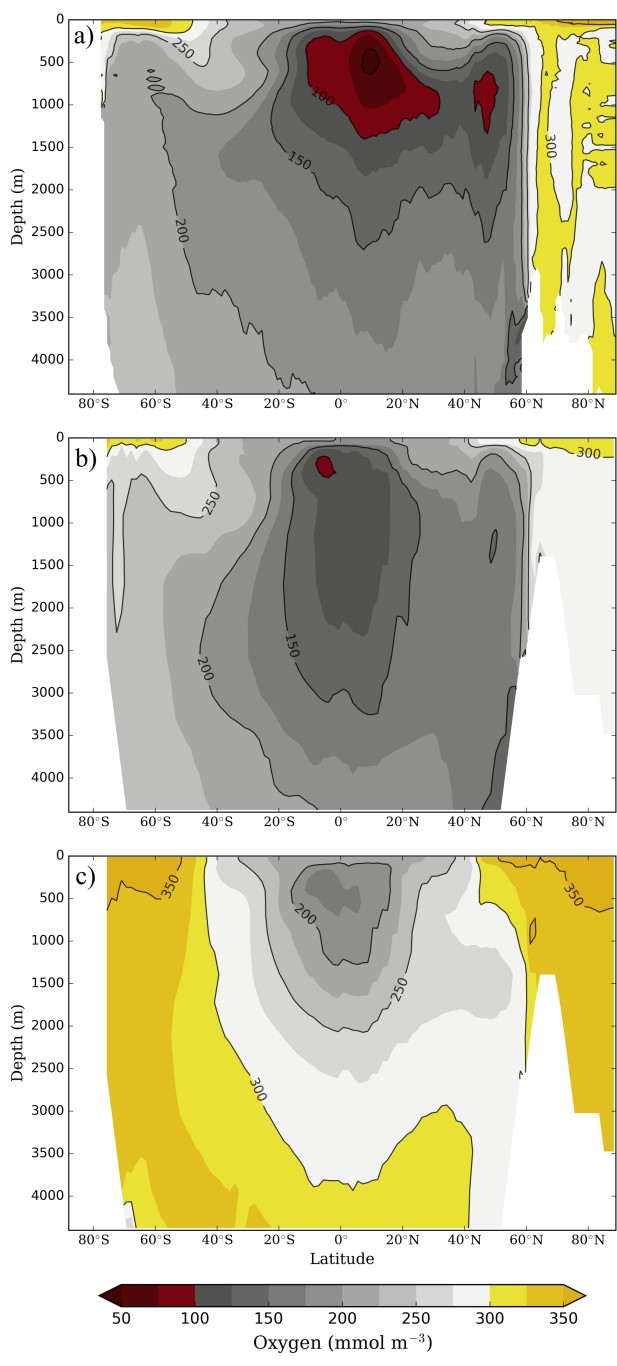

**Figure 9.** Zonally-averaged dissolved oxygen concentrations (mmol m$^{-3}$) in (a) the modern ocean according to the World Ocean Atlas (Garcia et al., 2013), (b) the O-PI1 experiment, and (c) the O-LGM1 experiment.



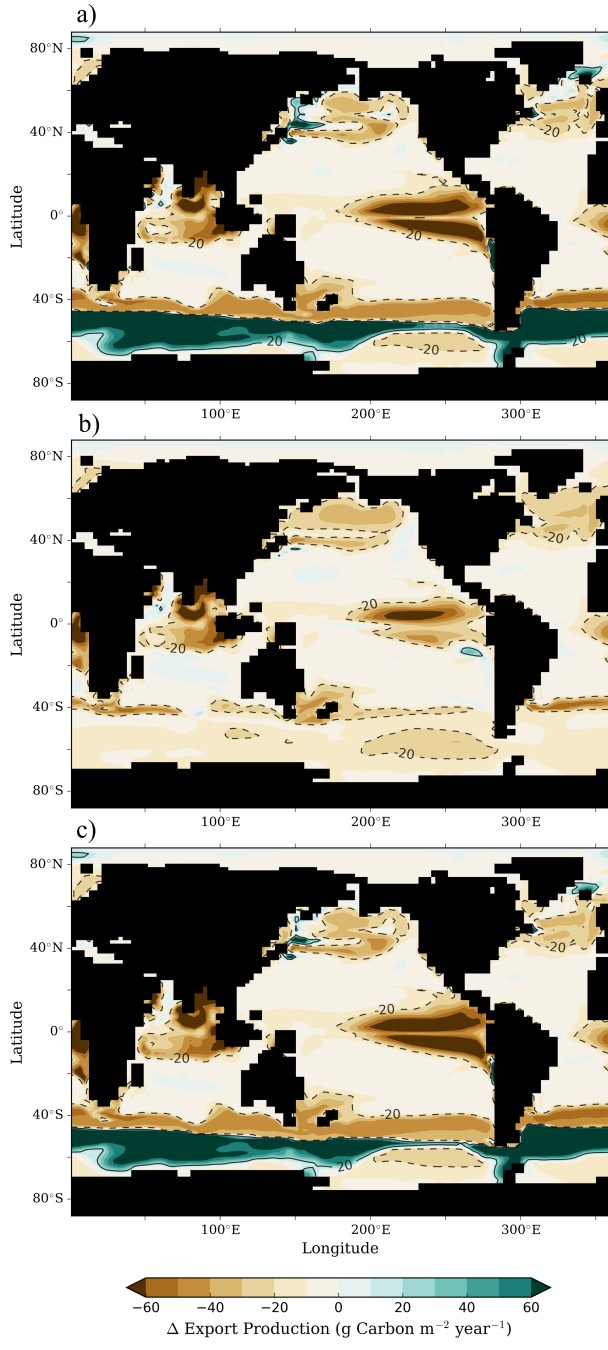

**Figure 10.** Change in annually averaged export of particulate organic carbon from the upper 50 m (g Carbon m$^{-2}$ year$^{-1}$) between the LGM and PI from the experiments with modified biogeochemical formulations for (a) O-LGM3 − O-PI1, (b) O-LGM4 − O-PI1, and (c) O-LGM6 − O-PI1. It should be noted that the export production field of particulate organic carbon for experiment O-LGM5, whereby particulate inorganic carbon was set to zero, did not differ from unmodified experiment O-LGM1 and is therefore not shown. For this comparison, the reader is directed to Figure 5.





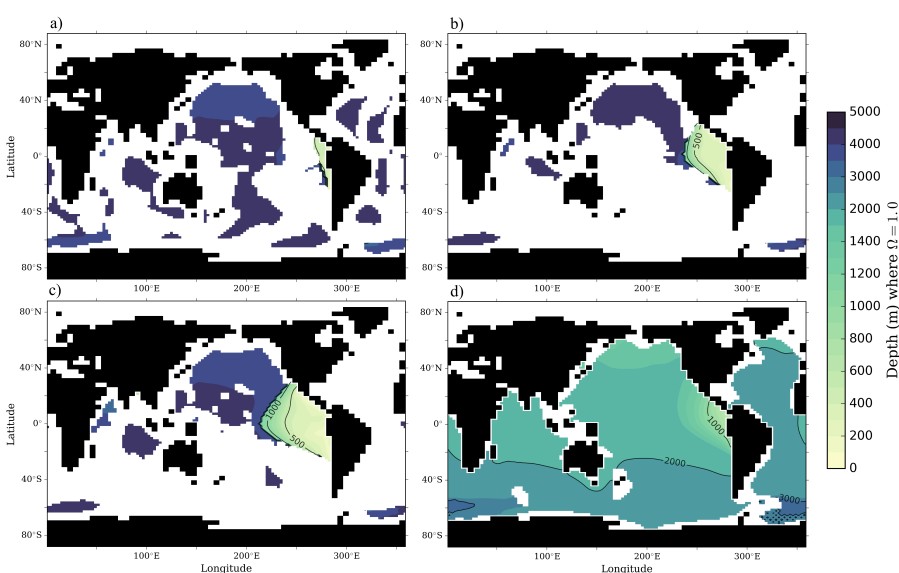

**Figure 11.** The depth of the aragonite saturation horizon ($\Omega = 1$) for the experiments with modified biogeochemical formulations. (a) O-LGM3, (b) O-LGM4, (c) O-LGM5, and (d) O-LGM6. The white areas in the ocean are regions where the aragonite saturate horizon is deeper than the ocean bottom.



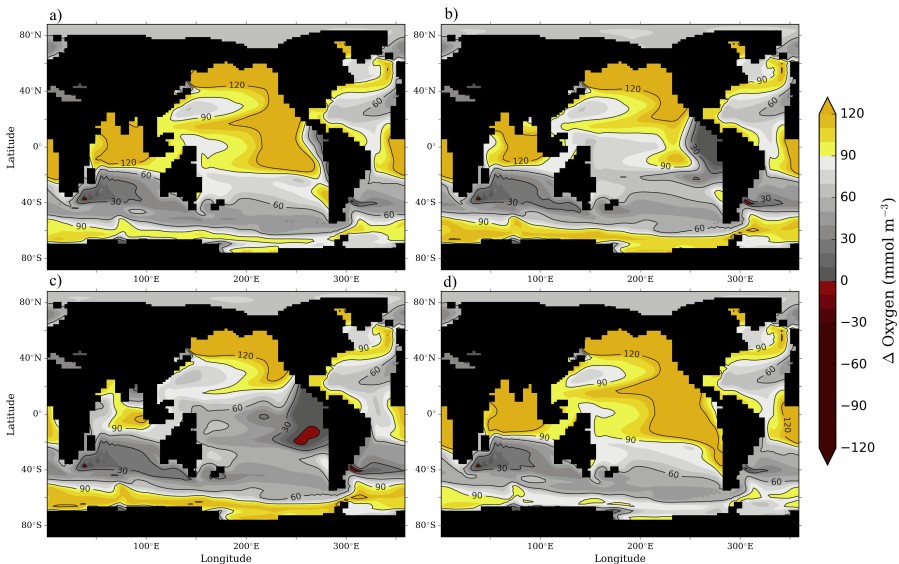

**Figure 12.** Change in oxygen concentration (mmol m$^{-3}$) at 500 m between the LGM and PI for the experiments with modified biogeo-chemical formulations. (a) O-LGM3 − O-PI1, (b) O-LGM4 − O-PI1, (c) O-LGM5 − O-PI1, and (d) O-LGM6 − O-PI1. A depth of 500 m is representative of the depth at which the greatest extent of low oxygen water exists in the simulated PI climate. It should be noted that the oxygen field for experiment O-LGM5, whereby particulate inorganic carbon was set to zero, did not differ from the unmodified glacial experiment O-LGM1 and can therefore be used here as a reference to that simulation.



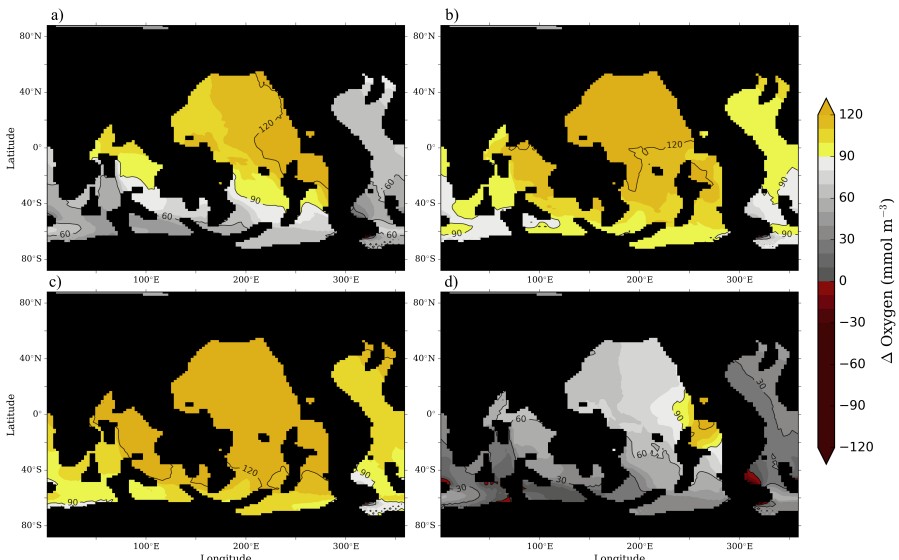

**Figure 13.** Change in oxygen concentration (mmol m$^{-3}$) at 3,500 m between the LGM and PI for the experiments with modified biogeo-chemical formulations. (a) O-LGM3 − O-PI1, (b) O-LGM4 − O-PI1, (c) O-LGM5 − O-PI1, and (d) O-LGM6 − O-PI1. A depth of 3,500 m is representative of the deep ocean. It should be noted that the oxygen field for experiment O-LGM5, whereby particulate inorganic carbon was set to zero, did not differ from the unmodified glacial experiment O-LGM1 and can therefore be used here as a reference to that simulation.