# Peer review of "The simulated climate of the Last Glacial Maximum and insights into the global marine carbon cycle"

_Climate of the Past, 2016_

## Referee Comment (RC1) · K. Matsumoto (Referee) · 10 Aug 2016

The authors tackle a grand old problem in geochemistry, the low glacial atmospheric pCO2. The topic is of interest to and appropriate for the CP readership.

Since Broecker's work in the 1980s, it's been well recognized that the ocean chemistry must have played a significant role. However, the community has struggled to explain the problem with any single mechanism or a combination/sequence of multiple mechanisms. This new modeling work by Buchanan does not offer a new mechanism per se but very nicely puts some of the major mechanisms within a theoretical framework offered by a single model architecture.

There is a lot to like in this submission. The coupled model simulations of physics (ice, circulation) are quite reasonable for both PI and LGM. I was impressed with their LGM

simulations that realized a deep chemical divide that separated the upper ocean (well ventilated, low values of C, P, ALK) from the deeper ocean. The rationale for requiring biogeochemical mechanisms is well articulated. There is good discussion of the new results with available paleoproxies. Data-model disagreement (e.g., deep $O_2$) and model shortcomings (e.g., inability to simulate open system carbonate compensation) are plainly presented. The paper is organized logically and well written. I believe this submission would make a nice contribution to the field and I support its publication.

I do have some comments and suggestions in no particular order that the authors may consider in their revision.

1) Overall I see newer papers being cited. While this shows that the authors are up to date and is obviously good, I also feel that some of the original papers should be cited. For example, David Archer has written many important papers on G-I CO2. There is a lot in his 2000 paper in Rev. of Geophysics. The lead author should read it, if he has not already, and cite it. Archer has explicitly modeled open system carbonate compensation in an OGCM, something that this work does not do and speculates on. Another example is that Broecker (1982) is probably one of the first to say that the ocean is key to the low glacial CO2, way earlier than the 2014-2015 papers cited (line 9, p. 2). There are other examples. My preference is that original papers are cited instead of newer papers that regurgitate original ideas.

2) The names of LGM experiments are just numeric (esp. 3-6) which I found a bit difficult to commit to my memory. Perhaps rename them to something more obviously descriptive.

3) The biogeochemical runs (LGM3-5) all simulate the effects of the desired modifications without explicitly modeling the modifications themselves mechanistically. The authors should discuss the actual mechanism (e.g., how do you envision the ocean actually increasing production? Fe? Excess nutrients from shelf weathering? How do you envision remineralization depth scale actually increasing? Temperature dependence?

But you actually only change the power law exponent everywhere without regard to how the temperature distribution changed. How would the ocean turn off PIC export? Diatom dominance due to Si leakage? What about corals?). The authors should then discuss possible impacts of the simplifications made.

4) Need reference for why whaling records give true estimates of ice.

5) Change "whom" to "who" on page 7, line 34.

6) Section 3.2.1 Carbon: I might suggest splitting the attribution of changes in carbon content to ocean physics into those driven by solubility, ice, and circulation. For example, how can we make better sense of the 517 PgC change (page 9, line 12)? You could create a mask for SST as it relates to solubility and control for that. Likewise, you could create a sea ice mask as it relates to gas exchange and control for that. Then you can get the circulation component by subtraction: total change – change due to solubility – change due to ice. An example of how you can do this splitting can be seen, for example, in Matsumoto et al. (2010) in Tellus. This splitting would require making masks from the PI1 run and doing additional experiments. . .extra work for sure.

7) What specifically about the glacial BCs that causes the NADW to go down by 25% and the AABW-NADW boundary to shoal by 1500 m? I think this is rather important to note.

8) Figure 1b: why is the Arctic warmer in LGM than PI?

9) The authors exclusively discuss aragonite when considering carbonate ion saturation. My sense is that it is far more common to discuss calcite over aragonite in the paleo literature as it related to the glacial CO2 problem. Lysocline is typically estimated based on forams in sediments. I would suggest that the authors switch to calcite in their revision.

10) Define omega on page 10, line 27.

11) It is incorrect to equate aragonite saturation horizon and the lysocline on page 11,

line 5-6. The former is a water column chemistry feature; the latter is a sedimentary feature. Also, the phrase "aragonite saturation horizon" sounds incorrect to me. It's more accurate to say, "carbonate ion saturation horizon with respective mineral aragonite." May be shorten the phrase after introducing it in full and correctly the first time it appears in the text. Again, it's far more common to talk about calcite than aragonite. It should be noted also that the lysocline and the saturation horizon (for whichever form of CaCO3 mineral) could theoretically be decoupled.

12) Despite the qualifier on page 13, line 6, my sense is that 326 Pg is still low. Glacial atmospheric CO2 was lower by ∼100 ppm (or ∼200 PgC). That only leaves 126 Pg (326-200) for change in terrestrial biosphere change. That seems too small. The deep ocean carbon isotope constraint on change in terrestrial biosphere (e.g., Shackleton, 1977) is still pretty strong in my view.

13) There must be a mistake on page 8, line 9. Can't be tens of thousands of Sv.

Enjoyed reading this.
* * *

---

## Referee Comment (RC2) · A. Schmittner (Referee) · 17 Aug 2016

Review of Buchanan et al. 2016 Climate of the Past Discussions: "The simulated climate of the Last Glacial Maximum and insights into the global carbon cycle"

The authors present model simulations of the LGM ocean biogeochemistry. I think the results are well described and make sense and the paper is well written and illustrated. The authors also compare their model to paleo reconstructions at least qualitatively. I agree with many of the authors findings but I think that some conclusions drawn need to be rephrased because the evidence provided is insufficient in supporting them.

The main issue I have with this paper is that the authors conclude both in the abstract (lines 13-14) and in the conclusion section (page 16, lines 8-9) that "physical changes . . . are not sufficient to explain " the $CO_2$ drawdown. I think these conclusions refer

to the physical changes simulated by their model. I don't think the authors can conclude that the model exactly reproduces the real LGM ocean physics. Therefore the statements should make clear that they do not refer to the real ocean but to the model simulated ocean. I suggest to rephrase by e.g. including in the abstract "physical changes simulated by our model cannot in isolation produce . . ." and a similar change in the conclusion section.

Below I also suggest to include some recent relevant references.

Page 2, line 5: include Annan and Hargreaves (2014; doi:10.5194/cp-9-367-2013) for a more up-to-date global mean estimate

Page 2, line 19: "inseparable" is a too strong word. I'd suggest "connected" instead.

Page 2, lines 20-24: In this discussion of physical mechanisms I would suggest to include recent work that has improved understanding of the effects of wind stress changes in the North Atlantic (Muglia and Schmittner, 2015, GRL, doi:10.1002/2015GL064583) and tidal mixing (Schmittner et al. 2015, GRL, doi: 10.1002/2015GL063561) on the circulation

Page 3, lines 32-33: please clarify if Bering Strait is open or closed?

Page 4 lines 4-6: I don't understand this sentence. How does coupling between a cooler atmosphere and the ocean lead to a 0.5 psu increase in salinity? That is about half the increase it should be based on the 120 m sea level drop.

Page 4, line 16: "averages" do you mean monthly averages or annual?

Page 4, line 30: please explain the other variables in this equation. What is F(I) ? Light limitation? Why is there a multiplication by 12? What is $V_{max}$ and $P_k$?

Page 7, lines 10-13: Note that Ferrari et al. (2014) do not consider other processes that we know are important for determining the MOC strength such as closure of Bering Strait (Hu et al. 2010, Nat. Geosc., DOI: 10.1038/NGEO729), wind stress and tidal

mixing changes (see above papers).

Page 7, line 33: " The Cpl-LGM sea ice in this study is broadly consistent with the palaeo evidence in the North Atlantic," the simulated perrenial sea ice cover seems inconsistent with proxy based evidence of seasonaly ice free Nordic Seas

Page 8, line 15: "18392 to 5391 Sv" looks like a typo

Page 8, line 16: See Muglia and Schmittner (2015) for updated numbers from the PMIP3 models

Page 8, line 22: check Broecker 2013 reference. I couldn't find it based on the information in the reference list.

Page 8, line 26: for a different view on diapycnal mixing see Schmittner et al. (2015) and references therein.

Page 8, lines 32-35: see my previous comments on Ferrari et al.

Page 10, lines 8-11: compare with data constrained model of Schmittner and Somes (2016, PO, doi: 10.1002/2015PA002905)

Page 11, lines 9-10: this is probably due to the neglect of sediment interactions and whole ocean alkalinity changes (see also discussion in Schmittner and Somes 2016)

Page 13, lines 29-31: I don't think that the reduction in export production is contrary to arguments for a strengthened biological pump. The strenth or efficiency of the biological pump is best defined as the global mean respired carbon or phosphorous content (see discussion in Schmittner and Somes 2016). Thus a more efficient biological pump which results in sequestration of organic carbon and nutrients in the deep ocean will cause an reduction in export production due to the loss of upper ocean nutrients.
* * *

---

## Author Comment (AC1) · 13 Oct 2016

In response to Katsumi Matsumoto:

1) The reviewer has made an important point here that we will rectify to the best of our ability. We agree completely with the reviewer in that it is important to pay homage to those scientists that made the initial connections, as well as referencing contemporary findings. We would hope to make the changes in the Introduction and there include some of the seminal studies that the review mentions, plus others that made important contributions to the field. However, it should be of note that we have payed homage already to the following seminal studies:

- The Broecker (1982) paper that the reviewer mentioned is indeed included in the text (p2, line 28), when we make a special effort to discuss the potential of ocean biology

for affecting the climate.

- Papers by Shackleton (1967) and Emiliani (1966) that were among the first to identify the sawtooth cycling between glacial-interglacial climates

- Jouzel (1987) and Petit (1999) discovered the remarkable correlation between atmospheric $CO_2$ and climate cycles

- Duplessey (1988) found that a chemical divide between the upper and deep ocean occurred during glacial periods, indicating reorganisations in the ocean circulation.

- Archer & Maier-Reimer (1994) was the first to use a model with carbon compensation to show that changes in the dissolution of calcium carbonate in the deep ocean can explain large changes in glacial-interglacial $CO_2$ changes.

2) The names of the experiments may be altered to reflect the changes that were made. The changes we might suggest are as follows:

- O-PI1 to O-PI

- O-PI2 to O-PI(pCO2-185)

- O-LGM1 to O-LGM

- O-LGM2 to O-LGM(pCO2-280)

- O-LGM3 to O-LGM(bgc-poc)

- O-LGM4 to O-LGM(bgc-rem)

- O-LGM5 to O-LGM(bgc-pic)

- O-LGM6 to O-LGM(bgc-all)

3) While justifications for the biogeochemical modifications have already been made in the Methods section, we can acknowledge that perhaps the prescription of changes (as opposed to their generation through modelling their mechanistic behaviour) was

not made clear enough in the text. To make this clear, we suggest adding a new paragraph in the Methods Section beneath line 10 on page 5: "These biogeochemical modifications are designed to capture fundamental changes in the oceanic biological pump that are likely to have occurred under glacial conditions (see above). They do not explicitly capture the modifications themselves in any mechanistic sense. Deepening the global remineralisation profile in experiment O-LGM$^{bgc}_{rem}$, for instance, is a prescribed change that is informed by global cooling, rather than a change caused by cooling that is simulated in the model. However, the prescription of these changes allows us to undertake a theoretical investigation into their capacity to sequester carbon at the LGM."

4) De La Mare (1997) Nature

5) Easily changed and will do so.

6) For this comment, the reviewer asks for additional experiments to be run. These experiments are currently underway and their results can be supplemented into the paper within a matter of days. This would, however, require an extra paragraph of text and additions to the major tables.

7) The increase in salinity throughout the global ocean had a greater effect on the formation of deep waters in the Southern Ocean than in other regions. In fact, sea surface salinity decreased over much of the low to mid latitude oceans, including the North Atlantic, while it increased in the Southern Ocean. Please see the supplemental figure that is attached, showing pre-industrial salinity field (left) and the LGM-PI difference (right) in psu.

8) The Arctic in general was not necessarily warmer. The confusion may have arisen due to the pink colour, but if special attention is given to the colour bar at the bottom of Figure 1, it can be seen that this pink colour centres around a 0°C change. However, the reviewer is right to point out that there is a small region of slightly warmer temperature in the Arctic above eastern Europe. On further investigation of our simulations,

the warmer spot as seen in the annual average sea surface temperature difference (Fig 1b) is a symptom of warmer late winter, spring and early summer temperatures. The warmer temperatures also caused lower sea ice coverage in this region during spring/early-summer, despite increased sea ice coverage in general throughout the NH. The increase in temperature and loss of sea ice during the spring in the surface waters in this region was caused by significantly reduced wind stresses that reduced heat fluxes out of the ocean in late winter and early spring. Because this is a relatively small result in the context of this study, we have not included the above discussion within the paper, which we feel would de-rail the logical flow of the text.

9) The reviewer makes a good point. We referred to aragonite primarily because, as the more unstable species of carbonate, it shows the greatest sensitivity to changes in alkalinity. Also, changes in surface aragonite are more appropriate when assessing the model against reconstructions of coral reef changes. We agree with the reviewer and suggest changing the manuscript to accommodate changes in the calcite saturation depth, rather than the aragonite saturation depth. This would mean altering:

- Paragraph 3 of section 3.2.3 (beginning page 10)

- Section 3.3.3 (beginning page 14)

However, we have not altered our discussion of the aragonite saturation state in the surface ocean, because this provides the best model-data comparison for coral reef reconstructions.

10) We can add an extra sentence to introduce what the aragonite/calcite saturation state is.

11) By following the reviewer's suggestion (9), this comment would be addressed.

12) While low, the 326 PgC that was added to the ocean does fall within the bounds of error provided by Ciais et al (2011). This is an important point. However, the reviewer is correct in pointing out that this number is on the lower bound of the estimates. Combined with the fact that the biogeochemical modifications could not deoxygenate the deep ocean, this indicates that additional processes (physical or biogeochemical) were not captured in the simulations, and therefore we would aim to make suggestions for future work to address these inconsistencies. This, we believe, is one of the critical conclusions of the study.

13) The total volume transport out of the Southern Ocean was calculated by taking the depth and longitudinal integrated transport (Sv) across 45°S to obtain in m yr-1. Thus, the reviewer is correct in that it is not Sverdrups and this is a mistake. However, for the sake of reader comprehension, we will alter these measurements to be in Sv and to reflect the average export out of the Southern Ocean across 45°S and in depth.

————————————————————

[Figure]

**Fig. 1.**

---

## Author Comment (AC2) · 13 Oct 2016

On behalf of all the authors for this study, we would like to thank Katsumi for his review of the paper. His comments will improve the quality of the manuscript.

---

## Author Comment (AC3) · 14 Oct 2016

The major suggestion by the reviewer is for a general re-wording of our conclusions. When we suggest that "physical changes cannot in isolation explain the necessary drawdown of CO2 at the Last Glacial Maximum", the reviewer asks for "the simulated physical changes cannot in isolation explain the necessary drawdown of CO2 at the Last Glacial Maximum". While this necessitates only a small number of additions to the paper, the effect on our conclusions is a significant one. However, we agree with the reviewer in their suggestion because we acknowledge that probably the biggest assumption of this work is that the physics of the LGM as simulated by the CSIRO Mk3L were an accurate representation of reality. Thus, we would undertake the re-wording as is suggested by Andreas Schmittner and feel that this change will significantly improve the manuscript.

The remaining comments by the reviewer are very specific in nature. These are suggestions for additional references, clarifications of model architecture/omissions, experimental caveats, and grammatical issues. We will to the best of our ability accommodate these changes into the manuscript.

We detail our responses to each specific suggestion made by the review below. These responses are made in the order to which the reviewer made their comments.

1) The work of Annan and Hargreaves (2013; doi:10.5194/cp-9-367-2013), has already been referenced in the manuscript in the section on Sea Surface Temperature (section 3.1.1).

2) The effects of wind stresses and tidal mixing can be added to the discussion of physical factors affecting the glacial sequestration of carbon and circulation changes.

3) The Bering Strait is open in the model, and we can make this point clear by adding an extra sentence.

4) The addition of 0.5 psu of salinity is indeed about half that estimated for a 120 m drop in sea level at the LGM. The accumulation of water in snow across the land caused the development of a drier atmosphere in the simulated LGM, and this increased the salinity of the ocean by increasing evaporation. Because we did not artificially add salinity to the ocean in the ocean-only experiments, the addition of salinity was maintained at 0.5 psu. It is possible for us to re-run our experiments by adding an additional 0.5 psu to the salinity field. This would alter our results, potentially causing an increased sequestration of carbon in the deep ocean by further increasing the salinity-driven density gradients.

5) Monthly averages. This can be added to the sentence.

6) We have neglected to include certain parts of the equation because they are available in Appendix A of Matear and Lention (2014), which we point the reader to for further information. However, we could easily make these additions to the methods

section. These additions would include:

- An explanation of the Michaelis-Menten relationship between nutrient availability and phytoplankton uptake.

- An explanation of the maximum growth rate of phytoplankton as dependent on temperature.

- An explanation of the light limitation term F(I).

The multiplication by 12 is to convert from moles carbon to grams of carbon, but this may not be necessary to include in the equation.

7) A disclaimer can be added to ensure that these processes are acknowledged and that the ready is clear that we do not consider them in this study.

8) It should be noted that we use the term "broadly" in this sentence. This refers to the overall pattern of expansion, including the greater increase in sea ice cover in the western North Atlantic relative to the east. Also, the sea ice cover is represented in the paper as a fractional cover, that is on a scale from 0 (no cover) to 1 (complete cover). Thus, seasonally-free ice cover in the eastern Nordic seas as suggested by proxy data (de Vernal et al, 2005) cannot distinguish if small amounts of sea ice was still present during annual minima. However, the reviewer makes a good point, as the conditions at the LGM in the North Atlantic most likely consisted of greater ice cover in the western Nordic seas and sea ice free summers in the eastern Nordic Seas.

This inconsistency can be addressed in the manuscript. We suggest that the sentence "this study is broadly consistent with the palaeo evidence in the North Atlantic" is kept, but that we acknowledge that the seasonal opening of the eastern Nordic Seas was not captured in the sentences prior.

However, it should be remembered that this is a course resolution climate system model and it cannot be expected to capture fine detail changes in sea ice within a region like the Nordic Seas. In fact, on closer inspection, large summertime reductions

in sea ice were simulated in the region north of Eastern Europe. Although this is not the eastern Nordic Seas, we again suggest that the model "broadly" captures the pattern of sea ice changes at the LGM in the North Atlantic.

9) This was indeed a mistake. The total volume transport out of the Southern Ocean was calculated by taking the depth and longitudinal integrated transport (Sv) across 45°S to obtain in m yr-1. Thus, the reviewer is correct in that it is not Sverdrups and this is a mistake. However, for the sake of reader comprehension, we will alter these measurements to be in Sv and to reflect the average export out of the Southern Ocean across 45°S and in depth.

10) We can update our model comparison values using the study that the reviewer proposes. This will necessitate an addition to Table 2 and will necessitate changes to our discussion of the changes in circulation. Namely, that the weakening of the AMOC is not consistent with PMIP3 simulations of the LGM conditions. This weakening in the AMOC is due to a reduction in North Atlantic salinity in the LGM simulation. SEE ATTACHED FIGURES Once again, this inconsistency may be rectified by re-completing the experiments with an artificial addition of 0.5 psu salt to the salinity field to ensure that the ocean increased in total salinity by 1 psu, consistent with a sea level drop of 120 m.

11) Follow this link –> http://www.ldeo.columbia.edu/∼broecker/Home_files/WhatDrvsGlacCycl.2.4.pdf

12) We agree that the increase in diapycnal mixing due to enhanced tidal mixing proposed by Schmittner et al (2015) should be acknowledged in the text. This can be added to this discussion, acknowledging that we do not consider tidal mixing in the coarse resolution climate model.

13) A comparison of our results with Schmittner and Somes (2016) will be a constructive addition to the manuscript and we thank the reviewer for bringing it to our attention. We aim to include all necessary comparisons, which will include direct comparisons between their export production fields and the carbon sequestration they achieve.

14) We have addressed the changes in the lysocline seen in the model experiments in light of our neglect of sediment interactions later in the manuscript. See section 3.3.3. Changes will also be made to this section based on the suggestions of the other reviewer.

15) We agree with the reviewer and expect to make the comparison with Schmittner and Somes (2016) regarding biological pump efficiency, relative to biological pump strength.
* * *
[Figure]

[Figure]

**Fig. 1.**

[Figure]

[Figure]

**Fig. 2.**

---

## Author Comment (AC4) · 14 Oct 2016

The suggestions made by Andreas Schmittner were highly poignant to the experiments that were conducted and will benefit the manuscript and its readability for a more technical audience.

---

## Author Response (AR1)

**Editor Decision: Publish subject to minor revisions (review by Editor)** (03 Nov 2016) by Prof. Arne Winguth
Comments to the Author:
Dear Dr. Buchanan,

Thank you for submitting your manuscript entitled "The simulated climate of the Last Glacial Maximum and insights into the global carbon cycle" [Paper #cp-2016-73] to Climate of the Past. I have now received an assessment by two reviewers who found moderate issues that will need to be addressed before we can finally accept the paper. The first reviewer suggested an additional simulation to evaluate the attribution of changes in the carbon content and I agree with him that this would make the paper stronger. I also noticed that the paper misses a discussion about feedbacks associated with marine sediments (as e.g. discussed in Archer et al., Rev. Geophysics, 2000) which should be included in the revised version.
I suggest to change the title of the paper to "The simulated climate of the Last Glacial Maximum and insights into the global marine carbon cycle" since it focuses on the marine carbon cycle.

The feedback to your manuscript provided by the reviewer assessments is important and should be taken into account as you complete your revision. Please include a point-by-point reply to the reviewer comments, consider the reviewers' concerns in the revision, and submit a marked-up manuscript version showing the changes made in your revision. I encourage you to submit a suitably revised version of your manuscript, if possible, by November 25, 2016.

Sincerely,
Arne Winguth
Editor

| Editor comment | Author response | Altered? |
|---|---|---|
| 1) The first reviewer suggested an additional simulation to evaluate the attribution of changes in the carbon content and I agree with him that this would make the paper stronger. | We have run two additional experiments to address this suggestions:
1. O-PI$^{LGM}_{-ice}$
2. O-PI$^{LGM}_{-sol}$

These experiments both use the pre-industrial overturning circulation, but prescribe atmospheric $CO_2$ at 185 ppm. For O-PI$^{LGM}_{-ICE}$, the sea ice field from the LGM climate has been prescribed and affects only the biogeochemical tracers. For O-PI$^{LGM}_{-sol}$, the sea surface temperature and salinity fields of the LGM climate are used to govern air-sea exchange of biogeochemical tracers.

The findings from these experiments have been added to the paper. These were that solubility can account for an additional storage of 349 Pg C, while sea ice expansion reduced the oceans carbon content by 160 Pg C. Thus, by the difference of 668 Pg C between the O-PI$^{LGM}_{CO2}$ and O-LGM experiments, we find that the circulation changes of the LGM | Y |

| | ocean caused an addition 479 Pg C to be held within the ocean. This confirmed the strong effect that circulation changes had on the carbon cycle at the LGM. | |
|---|---|---|
| 2) I also noticed that the paper misses a discussion about the feedbacks associated with marine sediments (as e.g. discussed in Archer et al. Rev. Geophysics, 2000) which should be included in the revised version. | We have discussed the limitations of not including sedimentary processes in section 3.3.3 (Reconciling carbonate chemistry). | Y |
| 3) I suggest to change the title of the paper to "The simulated climate of the Last Glacial Maximum and insights into the global marine carbon cycle" since it focuses on the marine carbon cycle. | The title has been changed to the editor's suggestion. | Y |

Additionally, we would like to alert the editor to a number of additional, minor changes that have been made to the manuscript. These are detailed in the track changes document that was uploaded with the revised version.

**Katsumi Matsumoto**

The authors tackle a grand old problem in geochemistry, the low glacial atmospheric pCO2. The topic is of interest to and appropriate for the CP readership. Since Broecker's work in the 1980s, it's been well recognized that the ocean chemistry must have played a significant role. However, the community has struggled to explain the problem with any single mechanism or a combination/sequence of multiple mechanisms. This new modeling work by Buchanan does not offer a new mechanism per se but very nicely puts some of the major mechanisms within a theoretical framework offered by a single model architecture.

There is a lot to like in this submission. The coupled model simulations of physics (ice, circulation) are quite reasonable for both PI and LGM. I was impressed with their LGM simulations that realized a deep chemical divide that separated the upper ocean (well ventilated, low values of C, P, ALK) from the deeper ocean. The rationale for requiring biogeochemical mechanisms is well articulated. There is good discussion of the new results with available paleoproxies. Data-model disagreement (e.g., deep O2) and model shortcomings (e.g., inability to simulate open system carbonate compensation) are plainly presented. The paper is organized logically and well written. I believe this submission would make a nice contribution to the field and I support its publication.

I do have some comments and suggestions in no particular order that the authors may consider in their revision.

| Reviewer comment | Author response | Altered? |
|---|---|---|
| 1) Overall I see newer papers being cited. While this shows that the authors are up to date and is obviously good, I also feel that some of the original papers should be cited. For example, David Archer has written many important papers on G-I CO2. There is a lot in his 2000 paper in Rev. of Geophysics. The lead author should read it, if he has not already, and cite it. Archer has explicitly modeled open system carbonate compensation in an OGCM, something that this work does not do and speculates on. Another example is that Broecker (1982) is probably one of the first to say that the ocean is key to the low glacial CO2, way earlier than the 2014-2015 papers cited (line 9, p. 2). There are other examples. My preference is that original papers are cited instead of newer papers that regurgitate original ideas. | The reviewer has made an important point here that we have rectified to the best of our ability. We agree completely with the reviewer in that it is important to cite original papers, as well as referencing contemporary findings. The changes have been made in the Introduction, and as such, the first paragraph of the introduction has seen some major changes.

We have cited the following seminal studies already:

1. The Broecker (1982) paper that the reviewer mentioned is indeed included in the text (p2, line 28), when we make a special effort to discuss the potential of ocean biology for affecting the climate.
2. Papers by Shackleton (1967) and Emiliani (1966) that were among the first to identify the sawtooth cycling between glacial-interglacial climates | Y |

| | | |
|---|---|---|
| | 3. Jouzel (1987) and Petit (1999) discovered the remarkable correlation between atmospheric $CO_2$ and climate cycles | |
| | 4. Duplessey (1988) found that a chemical divide between the upper and deep ocean occurred during glacial periods, indicating reorganisations in the ocean circulation. | |
| | 5. Archer & Maier-Reimer (1994) was the first to use a model with carbon compensation to show that changes in the dissolution of calcium carbonate in the deep ocean can explain large changes in glacial-interglacial $CO_2$ changes. | |
| | We have added the following studies: | |
| | 1. Shackleton (1977) showed that the terrestrial reservoir of carbon was diminished during glacial periods using changes in carbon isotopes. | |
| | 2. Sowers et al. (1991) and Broecker & Henderson (1998) show that $CO_2$ likely had a causal role in causing ice mass depletion during glacial terminations. | |
| | 3. Archer et al. (2000) has been added twice. First, to the exploration of nutrient increase in the glacial ocean (p2, line 28). Second, to the list of studies that explored changes in the calcium carbonate to organic carbon rain ratio (p3, line 10). | |
| | 4. Eppley (1972) has been added to refer to changes in microbial metabolism with temperature. | |
| 2) The names of LGM experiments are just numeric (esp. 3-6) which I found a bit difficult to commit to my memory. Perhaps rename them to something more obviously descriptive. | The names of the experiments have been altered to reflect the changes that were made. The changes are as follows:
• O-PI1 → O-PI
• O-PI2 → O-PI$^{LGM}_{CO2}$
• O-LGM1 → O-LGM
• O-LGM3 → O-LGM$^{BGC}_{-poc}$
• O-LGM4 → O-LGM$^{BGC}_{-rem}$
• O-LGM5 → O-LGM$^{BGC}_{-pic}$
• O-LGM6 → O-LGM$^{BGC}_{-all}$ | Y |

| | | Y/N |
|---|---|---|
| 3) The biogeochemical runs (LGM3-5) all simulate the effects of the desired modifications without explicitly modeling the modifications themselves mechanistically. The authors should discuss the actual mechanism (e.g., how do you envision the ocean actually increasing production? Fe? Excess nutrients from shelf weathering? How do you envision remineralization depth scale actually increasing? Temperature dependence? But you actually only change the power law exponent everywhere without regard to how the temperature distribution changed. How would the ocean turn off PIC export? Diatom dominance due to Si leakage? What about corals?). The authors should then discuss possible impacts of the simplifications made. | The mechanisms that inspired the biogeochemical modifications have been acknowledged in the text (beginning page 4, line 25).

For the experiment where export production is increased (O-LGM3), we justify this increase by evidence of enhanced iron fertilisation during the glacial period.
For the experiment where we increase the power law exponent (O-LGM4), we justify this global increase by evidence of reduced remineralisation rates in cooler waters.
For the experiment where we eliminate the production of inorganic carbon (O-LGM5), this is inspired by evidence that calcification is limited by temperature.

However, it may be that that the **prescription** of changes (as opposed to their generation through modelling their mechanistic behaviour) was not made clear enough in the text. To make this clear, we have added the following to the Methods Section:

It should be made clear that these experiments did not explicitly simulate the biogeochemical changes caused by an altered climate in any mechanistic sense. However, the prescription of the following changes allowed us to undertake a theoretical investigation into their capacity to sequester carbon at the LGM. | Y/N |
| 4) Need reference for why whaling records give true estimates of ice. | The sentence has been removed. | Y |
| 5) Change "whom" to "who" on page 7, line 34 | Changed | Y |
| 6) Section 3.2.1 Carbon: I might suggest splitting the attribution of changes in carbon content to ocean physics into those driven by solubility, ice, and circulation. For example, how can we make better sense of the 517 PgC change (page 9, line 12)? You could create a mask for SST as it relates to solubility and control for that. Likewise, you could | Two additional experiments were run:
1. O-PI$^{LGM}_{-Ice}$
2. O-PI$^{LGM}_{-sol}$

Experiment O-PI$_{LGM-Ice}$ isolated the effects of an expanded sea ice field on ocean carbon storage. Experiment O-PI$_{LGM-sol}$ isolated the effect of changes in sea | Y |

| | | |
|---|---|---|
| create a sea ice mask as it relates to gas exchange and control for that. Then you can get the circulation component by subtraction: total change – change due to solubility – change due to ice. An example of how you can do this splitting can be seen, for example, in Matsumoto et al. (2010) in Tellus. This splitting would require making masks from the PI1 run and doing additional experiments. . .extra work for sure. | surface temperature and salinity on ocean carbon storage.

The results of these experiments were integrated into section 3.2 (LGM climate: biogeochemical fields), and additional sentences were added to the methods.

With regard to the tables and figures, the experiments were added to Table 1, Table 3 and Table 4, as well as Figure 4. | |
| 7) What specifically about the glacial BCs that causes the NADW to go down by 25% and the AABW-NADW boundary to shoal by 1500 m? I think this is rather important to note. | The increase in salinity throughout the global ocean had a greater effect on the formation of deep waters in the Southern Ocean than in other regions. The surface density difference across the Southern Ocean increased (+0.9 kg m$^{-3}$), which increased the ACC transport (+160 Sv) and increased the subduction of AABW into the deep ocean.

We have added this discussion into the section concerning the Meridional Overturning Circulation. | Y |
| 8) Figure 1b: why is the Arctic warmer in LGM than PI? | The Arctic in general was not necessarily warmer. The confusion may have arisen due to the pink colour, but if special attention is given to the colour bar at the bottom of Figure 1, it can be seen that this pink colour centres around a 0°C change.

However, the reviewer is right to point out that there is a small region of slightly warmer temperature in the Arctic above eastern Europe. On further investigation of our simulations, the warmer spot as seen in the annual average sea surface temperature difference (Fig 1b) is a symptom of warmer late winter, spring and early summer temperatures. The warmer temperatures also caused lower sea ice coverage in this region during spring/early-summer, despite increased sea ice coverage in general throughout the NH.

The increase in temperature and loss of sea ice during the spring in the surface | N |

| | waters in this region was caused by significantly reduced wind stresses that reduced heat fluxes out of the ocean in late winter and early spring.

Because this is a relatively small result in the context of this study, we have not included the above discussion within the paper, which we feel would de-rail the logical flow of the text. | |
|---|---|---|
| 9) The authors exclusively discuss aragonite when considering carbonate ion saturation. My sense is that it is far more common to discuss calcite over aragonite in the paleo literature as it related to the glacial CO2 problem. Lysocline is typically estimated based on forams in sediments. I would suggest that the authors switch to calcite in their revision. | The reviewer makes a good point.

We referred to aragonite primarily because, as the more unstable species of carbonate, it shows the greatest sensitivity to changes in alkalinity. Also, changes in surface aragonite are more appropriate when assessing the model against reconstructions of coral reef changes.

We agree with the reviewer and have changed the manuscript to accommodate changes in calcite saturation, rather than aragonite saturation. As such, the following changes have been made:
  1. Paragraph 3 of section 3.2.3 (beginning page 10)
  2. Section 3.3.3 (beginning page 14)

However, we have not altered our discussion of the aragonite saturation state in the surface ocean, because this provides the best model-data comparison for coral reef reconstructions. | Y/N |
| 10) Define omega on page 10, line 27 | Defined.

… aragonite saturation state ($\Omega_{ar}$), which is a unitless index indicating under- and super-saturation at values below and above one (Fig. 7). | |
| 11) It is incorrect to equate aragonite saturation horizon and the lysocline on page 11, line 5-6. The former is a water column chemistry feature; the latter is a sedimentary feature. Also, the phrase "aragonite saturation horizon" sounds incorrect to me. It's more accurate to say, "carbonate ion saturation horizon with respective mineral aragonite." May be shorten the phrase after introducing it | We have referred to calcite saturation now in the revised version. All discussion of the lysocline is related to calcite and we have made this clear in the text. | Y |

| | | |
|---|---|---|
| in full and correctly the first time it appears in the text. Again, it's far more common to talk about calcite than aragonite. It should be noted also that the lysocline and the saturation horizon (for whichever form of $CaCO_3$ mineral) could theoretically be decoupled. | | |
| 12) Despite the qualifier on page 13, line 6, my sense is that 326 Pg is still low. Glacial atmospheric $CO_2$ was lower by ~100 ppm (or ~200 PgC). That only leaves 126 Pg (326-200) for change in terrestrial biosphere change. That seems too small. The deep ocean carbon isotope constraint on change in terrestrial biosphere (e.g., Shackleton, 1977) is still pretty strong in my view. | The ability of the model to simulate an increase in the carbon content of the ocean is made clear by our results, and our conclusion is that these biogeochemical and physical changes were sufficient to increase carbon sequestration to within the bounds of error around the estimated loss from the terrestrial and atmospheric reservoirs.

We also are clear in our concluding remarks that additional mechanisms must be missing from our simulations. In light of the palaeodata that we assess our simulations with, the additional mechanisms are most likely a further increase in the strength of the biological pump. This would not only increase carbon sequestration, thereby addressing the comment by the reviewer, but also further deoxygenate the deep ocean, thereby reconciling our most outstanding model-data difference. | Y |
| 13) There must be a mistake on page 8, line 9. Can't be tens of thousands of Sv | Corrected. | Y |

**Andreas Schmittner**

*Review of Buchanan et al. 2016 Climate of the Past Discussions: "The simulated climate of the Last Glacial Maximum and insights into the global carbon cycle".*

The authors present model simulations of the LGM ocean biogeochemistry. I think the results are well described and make sense and the paper is well written and illustrated. The authors also compare their model to paleo reconstructions at least qualitatively. I agree with many of the authors findings but I think that some conclusions drawn need to be rephrased because the evidence provided is insufficient in supporting them.

The main issue I have with this paper is that the authors conclude both in the abstract (lines 13-14) and in the conclusion section (page 16, lines 8-9) that "physical changes . . . are not sufficient to explain" the CO2 drawdown. I think these conclusions refer to the physical changes simulated by their model. I don't think the authors can conclude that the model exactly reproduces the real LGM ocean physics. Therefore, the statements should make clear that they do not refer to the real ocean but to the model simulated ocean. I suggest to rephrase by e.g. including in the abstract "physical changes simulated by our model cannot in isolation produce . . ." and a similar change in the conclusion section.

The major suggestion by the reviewer is for a general re-wording of our conclusions. When we suggest that "physical changes cannot in isolation explain the necessary drawdown of $CO_2$ at the Last Glacial Maximum", the reviewer asks for "the simulated physical changes cannot in isolation explain the necessary drawdown of $CO_2$ at the Last Glacial Maximum". While this necessitates only a small number of additions to the paper, the effect on our conclusions is a significant one. However, we agree with the reviewer in their suggestion because we acknowledge that probably the biggest assumption of this work is that the physics of the LGM as simulated by the CSIRO Mk3L were an accurate representation of reality. Thus, we re-worded the discussion as is suggested by the reviewer and feel that this change has significantly improved the manuscript.

Below I also suggest to include some recent relevant references:

| Reviewer comment | Author response | Altered? |
|---|---|---|
| 1) Page 2, line 5: include Annan and Hargreaves (2014; doi:10.5194/cp-9-367-2013) for a more up-to-date global mean estimate | The work of Annan and Hargreaves (2013; doi:10.5194/cp-9-367-2013), has already been referenced in the manuscript in the section on Sea Surface Temperature (section 3.1.1). | N |
| 2) Page 2, line 19: "inseparable" is a too strong word. I'd suggest "connected" instead. | Changed to connected | Y |
| 3) Page 2, lines 20-24: In this discussion of physical mechanisms I would suggest to include recent work that has improved understanding of the effects of wind stress changes in the North Atlantic (Muglia and Schmittner, 2015, GRL, | The Muglia & Schmittner (2015) paper refers to an increase in westerly wind stresses in the Northern Hemisphere among the PMIP3 LGM experiments. The increase in westerly winds is responsible for driving a stronger | Y |

| | | |
|---|---|---|
| doi:10.1002/2015GL064583) and tidal mixing (Schmittner et al. 2015, GRL, doi: 10.1002/2015GL063561) on the circulation | Atlantic Meridional Overturning Circulation (AMOC) than that of the Pre-Industrial circulation.

The Schmittner et al. (2015) paper refers to how increased tidal mixing during the LGM due to lowered sea level would have accelerated the overturning circulation.

However, both of these findings contrast with proxy reconstructions and other modelling studies of overturning circulation at the glacial maximum. The deep ocean, for instance, was almost universally deoxygenated. While we do not know for sure why the ocean was deoxygenated, it cannot co-occur with an intensified overturning circulation. The shoaling of the AMOC is clearly reconstructed by $\delta^{13}$C data. With regard to the modelling studies (including Schmittner & Somes, 2016, Paleoceanography), those that include biogeochemistry consistently require a shoaled AMOC to improve agreement between their simulated fields and proxy records.

We have included such a discussion, referring to the findings of Muglia & Schmittner (2015), within the section on Overturning Circulation. | |
| 4) Page 3, lines 32-33: please clarify if Bering Strait is open or closed? | The sentence "The closure of important oceanic connections due to sea level loss, such as the Bering Strait, was therefore not considered." was added. | Y |
| 5) Page 4 lines 4-6: I don't understand this sentence. How does coupling between a cooler atmosphere and the ocean lead to a 0.5 psu increase in salinity? That is about half the increase it should be based on the 120 m sea level drop. | The sentence was changed to be more clear: "Over this integration the ocean experienced an increase in salinity by 0.5 psu due to increased evaporation, which reflected the coupling between a cooler, drier atmosphere and the ocean."

To be clear, we therefore did not add salinity to the Cpl-LGM experiment, but it increased regardless due to model drift over the 5000 year simulation of the LGM climate. | Y |

| | We did investigate the impact of the salinity bias on the ocean circulation by correcting the SSS forcing and it did not alter the ocean dynamics. The simulations were re-completed with additional salt (making the total LGM-PI difference 1 psu) and the results of these simulations have now been used in the revised paper. | |
|---|---|---|
| 6) Page 4, line 16: "averages" do you mean monthly averages or annual? | Changed to monthly | Y |
| 7) Page 4, line 30: please explain the other variables in this equation. What is F(I) ? Light limitation? Why is there a multiplication by 12? What is V_max and P_k? | We have neglected to include certain parts of the equation because they are available in Appendix A of Matear and Lention (2014), which we point the reader to for further information.

However, we have included a very brief explanation of what the $V_{max}$, $P_k$ and F(I) terms represent in equation 1.

The multiplication by 12 is to convert from moles carbon to grams of carbon, but this is not necessary to include in the equation and we have removed it for conceptual clarity. | Y |
| 8) Page 7, lines 10-13: Note that Ferrari et al. (2014) do not consider other processes that we know are important for determining the MOC strength such as closure of Bering Strait (Hu et al. 2010, Nat. Geosc., DOI: 10.1038/NGEO729), wind stress and tidal mixing changes (see above papers). | This sentence has been removed. | Y |
| 9) Page 7, line 33: " The Cpl-LGM sea ice in this study is broadly consistent with the palaeo evidence in the North Atlantic," the simulated perrenial sea ice cover seems inconsistent with proxy based evidence of seasonaly ice free Nordic Seas. | It should be noted that we use the term "broadly" in this sentence. This refers to the overall pattern of expansion, including the greater increase in sea ice cover in the western North Atlantic relative to the east. However, the reviewer makes a good point, as the conditions at the LGM in the North Atlantic most likely consisted of greater ice cover in the western Nordic seas and sea ice free summers in the eastern Nordic Seas. This inconsistency between the model and proxies has been addressed. | Y |

| | It should be remembered that this is a course resolution climate system model and it cannot be expected to capture fine detail changes in sea ice within a region like the Nordic Seas. In fact, on closer inspection, large summertime reductions in sea ice were simulated in the region north of Eastern Europe. Although this is not the eastern Nordic Seas, we again would reinforce that the model "broadly" captures the pattern of sea ice changes at the LGM in the North Atlantic. | |
|---|---|---|
| 10) Page 8, line 15: "18392 to 5391 Sv" looks like a typo. | This has been removed and replaced with the rate of transport in the Antarctic Circumpolar Current as another metric of circulation change. | Y |
| 11) Page 8, line 16: See Muglia and Schmittner (2015) for updated numbers from the PMIP3 models. | Citation added, and the discussion around this result was modified.

The information contained by the results of Muglia and Schmittner (2015) significantly improved the interpretation of the results in this study. This citation alerted the authors to a number of recent insights into the AMOC during the glacial period (i.e. Howe et al. (2016) Nature Comms), which have put our results in a new light.

Consequently, we have added another paragraph to the discussion around the changes to the Meridional Overturning Circulation, which talks about our results in light of these recent discoveries.

While our conclusions remain the same, it does highlight important steps for improvement of model simulations of the LGM in years to come. | Y |
| 12) Page 8, line 22: check Broecker 2013 reference. I couldn't find it based on the information in the reference list. | Removed. | |

| | | |
|---|---|---|
| 13) Page 8, line 26: for a different view on diapycnal mixing see Schmittner et al. (2015) and references therein. | The connection between "reduced diapycnal mixing" and an increase in the dominance of AABW has been removed.

As such, diapycnal mixing due to interactions with bathymetry or tides is not discussed. We feel that this improves the discussion since we do not explicitly model these processes. | Y |
| 14) Page 8, lines 32-35: see my previous comments on Ferrari et al. | This section has been re-written, with a new discussion focusing on the reviewer's findings in their Muglia and Schmittner (2015) paper. This has improved the interpretation of our findings. | Y |
| 15) Page 10, lines 8-11: compare with data constrained model of Schmittner and Somes (2016, PO, doi: 10.1002/2015PA002905). | Included, in a qualitative manner, as Schmittner and Somes (2016) show a more efficient biological pump at the LGM in their simulations evidenced by increased regenerated carbon. | Y |
| 16) Page 11, lines 9-10: this is probably due to the neglect of sediment interactions and whole ocean alkalinity changes (see also discussion in Schmittner and Somes 2016). | We address this limitation further on in the discussion of biogeochemical changes. | N |
| 17) Page 13, lines 29-31: I don't think that the reduction in export production is contrary to arguments for a strengthened biological pump. The strenth or efficiency of the biological pump is best defined as the global mean respired carbon or phosphorous content (see discussion in Schmittner and Somes 2016). Thus a more efficient biological pump which results in sequestration of organic carbon and nutrients in the deep ocean will cause an reduction in export production due to the loss of upper ocean nutrients. | Our discussion has been altered to accommodate the suggestion of the reviewer. We point out that the biological pump was indeed weakened in our LGM simulations by reporting the loss in regenerated carbon, in spite of our BGC modifications. We then discuss this weakening in light of the evidence of an strengthened biological carbon pump (Galbraith et al. (2015) and Schmittner and Somes (2016)). This inconsistency, in combination with the inability of the simulations to deoxygenate the deep ocean, leads us to conclude that the strength of export production could be further increased. | Y |

[revised manuscript text omitted]
 | Oxygen (mean) (mmol m⁻³) Upper[e] | Oxygen (mean) (mmol m⁻³) Deep[f] | Depth[c] (m) where Ω = 1 |
|---|---|---|---|---|---|---|---|---|---|---|---|---|
| O-PI1 | 280 | 8.01 | 1.60 | 0.64 | 0 | 0 | 0 | 1650 | 181 | 182 | 180 | 1526 |
| O-PI2 | 185 | | | | −1486 | −801 | −686 | | | | | |
| O-LGM1 | 185 | 4.48 | 0.76 | 0.36 | −517 | −839 | 322 | 667 | 281 | 263 | 301 | 2944 |
| O-LGM2 | 280 | | | | 1127 | −40 | 1165 | | | | | |
| O-LGM3 | 185 | 5.92 | 4.06 | 0.47 | −329 | −933 | 603 | 772 | 272 | 271 | 273 | 2995 |
| O-LGM4 | 185 | 3.25 | 0.74 | 0.26 | −367 | −760 | 391 | 721 | 276 | 266 | 288 | 2994 |
| O-LGM5 | 185 | 4.48 | 0.76 | 0.00 | −255 | −447 | 190 | 667 | 281 | 263 | 301 | 2883 |
| O-LGM6 | 185 | 4.82 | 3.44 | 0.00 | 326 | −406 | 732 | 1004 | 252 | 265 | 238 | 1818 |

[a] Atmospheric CO₂ is prescribed in each experiment.

[revised manuscript text omitted]